# Decentralized Two-Sided Bandit Learning in Matching Market

**Yirui Zhang**[1]                    **Zhixuan Fang**[*1,2]

[1]Institute for Interdisciplinary Information Sciences., Tsinghua University, Beijing, China
[2]Shanghai Qi Zhi Institute, Shanghai, China,

## Abstract

Two-sided matching under uncertainty has recently drawn much attention due to its wide applications. Existing works in matching bandits mainly focus on the one-sided learning setting and design algorithms with the objective of converging to stable matching with low regret. In this paper, we consider the more general two-sided learning setting, i.e. participants on both sides have to learn their preferences over the other side through repeated interactions. Inspired by the classical result that the optimal matching for the proposing side can be obtained using the Gale-Shapley algorithm, our inquiry stems from the curiosity about whether this result still holds in a two-sided learning setting. To handle this question, we formally introduce the two-sided learning setting, addressing strategies for both the arm and player sides without restrictive assumptions such as special preference structure and observation of winning players. Our results not only provide a positive answer to our inquiry but also offer a near-optimal upper bound, achieving $O(\log T)$ regret.

## 1 INTRODUCTION

Stable matching with preferences on both sides is a classic problem with wide applications encompassing marriage, college admission, and labor markets. The classical literature [Roth and Sotomayor, 1992, Roth and Xing, 1997, Gale and Shapley, 1962] usually focuses on how to generate a stable outcome, i.e. how to find a stable matching where no pair wants to swap their partners. However, these works usually assume that every participant is aware of her own

preference perfectly beforehand, which may not be satisfied in many scenarios. As an illustration, consumers may lack knowledge about the service qualities offered by service providers, and workers may find themselves unaware of the value associated with the provided positions. In contrast to the assumption of perfect prior knowledge of individual preferences, participants in real-world scenarios usually acquire information about their own utilities through repeated interactions. For instance, in online crowdsourcing platforms (Upwork and TaskRabbit) and question-answering platforms (Quora, Stack Overflow), participants engage in repeated transactions, receiving stochastic rewards and learning their preferences over time. This uncertain aspect of preference acquisition introduces complexities that go beyond the traditional stable matching literature, forming the basis for exploration in more realistic matching scenarios.

Recent literature on matching bandits ([Liu et al., 2020, Basu et al., 2021, Liu et al., 2021, Sankararaman et al., 2021, Maheshwari et al., 2022]) has initiated exploration into scenarios where participants on one side seek to learn preferences through bandit feedback. We refer to this setting as the one-sided learning in matching bandits. In the context of learning uncertain individual preferences through repeated interactions, a pivotal question within the domain of matching markets revolves around understanding the convergence to equilibrium. In response to this question, many works in matching bandits propose algorithms with the objective of achieving stable matching with low regret. However, given the absence of prior information about individual preferences and a centralized platform for information collection, many works resort to assumptions to simplify the preference learning process. Some assume direct observation ([Kong and Li, 2023, Liu et al., 2021, Pokharel and Das, 2023]), while others leverage special preference structures ([Basu et al., 2021, Maheshwari et al., 2022, Sankararaman et al., 2021]).

In our work, we study the more general case where both sides lack the knowledge of their own preferences, referred to as the two-sided learning setting. Importantly, we do not

---

[*]Corresponding author: Zhixuan Fang at Tsinghua University (zfang@mail.tsinghua.edu.cn). This work is supported by Tsinghua University Dushi Program.

make assumptions regarding observations or impose special preference structures. Based on classical results in the matching market, the optimal matching for the proposing side can be obtained using the Gale-Shapley algorithm. Our inquiry revolves around the exploration of whether this result remains applicable in the context of a two-sided learning setting. Specifically, our study involves two distinct sides of participants: the player side and the arm side. At each time slot, players simultaneously propose to arms, and arms select one proposal from all the candidates. Every participant can learn her own preference only through the rewards obtained after each match. Our objective is to ascertain whether these participants can converge to the player-optimal stable matching and, if so, how fast the convergence occurs.

Intuitively thinking, in the two-sided learning setting, achieving a stable equilibrium appears improbable if arms consistently provide inaccurate feedback about their preferences. To avoid this situation, we make several reasonable assumptions. Given that players' decisions hinge on feedback from arms, the efficient learning of preferences by arms is crucial to providing valuable feedback to players. Therefore, the learning speed of arms is the key to the problem. We measure the learning difficulty of the arm side by comparing it with the player side. In the main part of our paper, we consider a plausible scenario where the difficulty level of arms' preferences learning is comparable with that of players, up to a constant ratio of $D$. Additionally, taking into account the rationality of arms—specifically, their intention to maximize utilities—we introduce the concept of the "rational condition" and delve into the scenario where arms' strategies satisfy this condition.

As for the player-side strategies, we propose a new algorithm for the complex two-sided learning setting and provide rigorous regret analysis. Our results show that the market converges to the optimal stable matching at a logarithmic rate. Specifically, our algorithm achieves $O(\log T/\Delta^2)$ regret with respect to player-optimal stable matching, where $T$ represents the time horizon and $\Delta$ represents the minimal gap of player utilities. This regret bound is tight in terms of $T$ and $\Delta$. The regret bound also matches with the state-of-art result in the simpler one-sided learning setting. Furthermore, the algorithm design and theoretical analysis methods themselves may also serve as a preliminary step for future studies in the two-sided learning matching bandits.

Moreover, as an extension and a preliminary investigation into the realm of more diverse strategies for arms, we consider the case where arms adopt strategic policies to collaborate with players without the assumption of learning difficulty.

## 1.1 RELATED WORK

MAB is a classic and well-studied framework that models the decision-making process under uncertainty ([Katehakis and Veinott Jr, 1987, Auer et al., 2002]). A player faces $K$ arms with different utilities and aims to find out the best arm based on the stochastic reward received after each pull. The explore-then-commit (ETC) methods ([Garivier et al., 2016, Rosenski et al., 2016]), UCB-based strategies ([Li et al., 2010]) and Bayesian-type policies ([Chapelle and Li, 2011, Scott, 2010]) are commonly used to address the trade-off between exploration and exploitation and to minimize regret.

The first work that combines MAB framework and matching markets is from [Das and Kamenica, 2005], and Das and Kamenica [2005] propose an algorithm with numerical study under the strong assumption that each side of the market is homogeneous. Liu et al. [2020] generalize the MAB based matching and propose basic ETC type and UCB type algorithms. However, Liu et al. [2020] mainly consider the centralized setting which is not so practical in reality.

Later, a line of research emerged to study decentralized matching bandits with one-sided learning. As mentioned earlier, various works make different assumptions about arm preferences. For instance, Sankararaman et al. [2021] analyze the scenario of globally ranked players where all arms rank players in the same order. Later, Basu et al. [2021] consider a more general case of uniqueness consistency and propose UCB-D4. Another specific case, $\alpha$-reducibility, is explored by Maheshwari et al. [2022]. These assumptions are designed to ensure a unique stable matching.

When examining general preferences without constraints, it is common for multiple stable matches to exist in the market. The least preferred stable matching for players is referred to as the player-pessimal stable matching, while the most preferred one is termed the player-optimal stable matching. Regret, defined concerning the optimal stable matching, is more desirable, as comparing it with the pessimal stable matching could result in additional linear regret compared to the optimal stable matching. With accurate knowledge of arm preferences on both the arm side and player side, Liu et al. [2021] design a conflict-avoiding algorithm named CA-UCB, which upper-bounds the player-pessimal stable regret under the assumption of "observation." Similarly, Kong et al. [2022] analyze a Thompson Sampling-based conflict-avoiding algorithm with "observation." Focusing on general preferences, Basu et al. [2021] propose a phased-based algorithm but with a high exponential dependency on $\frac{1}{\Delta}$. Adopting the assumption of "observation," Kong and Li [2023] propose ETGS, which guarantees player-optimal stable regret. ML-ETC, proposed by Zhang et al. [2022], is an ETC-based algorithm that can apply to general preference structures, and it also upper-bounds the player-optimal stable regret without "observation".

Table 1: Comparison between our work and prior results.

| | Assumptions | Player-Stable Regret |
|---|---|---|
| [Liu et al., 2020] | one-sided, centralized, known $\Delta$ | $O(K \log T / \Delta^2)*$ |
| [Liu et al., 2020] | one-sided, centralized | $O(NK^3 \log T / \Delta^2)$ |
| [Sankararaman et al., 2021] | one-sided, globally ranked | $O(NK \log T / \Delta^2)$ |
| [Basu et al., 2021] | one-sided, uniqueness consistency | $O(NK \log T / \Delta^2)$ |
| [Maheshwari et al., 2022] | one-sided, $\alpha$-reducibility | $O(CNK \log T / \Delta^2)$ |
| [Liu et al., 2021] | one-sided, observation | $O(\exp{(N^4)}K^2 \log^2 T / \Delta^2)$ |
| [Kong et al., 2022] | one-sided, observation | $O(\exp{(N^4)}K^2 \log^2 T / \Delta^2)$ |
| [Basu et al., 2021] | one-sided | $O(K \log^{1+\epsilon} T / \Delta^2 + \exp{(1/\Delta^2)})*$ |
| [Zhang et al., 2022] | one-sided | $O(K \log T / \Delta^2)*$ |
| [Kong and Li, 2023] | one-sided, observation | $O(K \log T / \Delta^2)*$ |
| [Pokharel and Das, 2023] | two-sided, observation | no theoretical results |
| [Pagare and Ghosh, 2023] | two-sided | $O(T_0 (K \log T / T_0 \Delta^2)^{1/\gamma} + T_0 (T/T_0)^\gamma)*$ |
| this paper | two-sided | $O(K \log T / \Delta^2)*$ |

[1] $K$ is the number of arms and $N$ is the number of players.

[2] $*$ represents the type of regret bound is player-optimal.

[3] $C$ relates to preferences and may grow exponentially in $N$.

[4] $\epsilon, \gamma$ are positive hyper-parameters, $\gamma$ belongs to $(0, 1)$. $T_0$ is a hyper-parameter that needs information about $\Delta$.

[5] The table categorizes "one-sided" versus "two-sided" based on whether learning involves interaction from both parties, rather than focusing on market characteristics.

The literature mentioned above often assumes knowledge of arm preferences and relies on precise feedback from arms. In contrast, Pokharel and Das [2023] address the scenario where preferences on both sides are unknown in matching bandits. They propose the PCA-DAA algorithm, incorporating random delays to reduce the likelihood of conflicts, though their findings are currently supported only by empirical results. In a recent study, Pagare and Ghosh [2023] introduce a multi-epoch ETC-type algorithm that achieves sub-linear regret. While the multi-epoch approach effectively reduces regret to a sub-linear level, it tends to induce over-exploration, resulting in still significant regret, typically polynomial. Their algorithm also necessitates the specification of a hyper-parameter $T_0$, which is constrained by requirements pertaining to knowledge of the minimal gap. Specifically, their approach relies on the assumption of a non-small gap. Additionally, the algorithm mandates that arms employ a symmetry algorithm similar to that used by players. Other works explore two-sided learning matching within the bandit framework from various angles. For example, Jagadeesan et al. [2023] delve into matching markets under the stochastic contextual bandit model, where a platform, at each round, selects a market outcome with the goal of minimizing cumulative instability.

## 2 MODEL

Suppose there are $N$ players and $K$ arms, and denote the set of players and arms by $\mathcal{N}$ and $\mathcal{K}$ respectively. We adopt the commonly used assumption in matching bandits that $N \leq K$ ( e.g. [Liu et al., 2021, Kong and Li, 2023, Basu

et al., 2021, Liu et al., 2020, Basu et al., 2021])[1]. Both the player side and arm side are unaware of their preferences. Specifically, for each player $j$, she has a fixed but unknown utility $u_{jk}$ associated with each arm $k$ and prefers arms with higher utilities. For each arm $k$, it also has a fixed but unknown utility $u_{kj}^a$ associated with each player $j$ and prefers players with higher utilities (the superscript $a$ stands for "arm"). Without loss of generality, we assume all utilities are within $[0, 1]$, i.e. for every $j \in \mathcal{N}, k \in \mathcal{K}$, $u_{jk}, u_{kj}^a \in [0, 1]$. Define the utility gap for player $j$ as $\Delta_j = \min_{k_1, k_2 \in \mathcal{K}, k_1 \neq k_2} |u_{jk_1} - u_{jk_2}|$ and the utility gap for arm $k$ as $\Delta_k^a = \min_{j_1, j_2 \in \mathcal{N}, j_1 \neq j_2} |u_{kj_1}^a - u_{kj_2}^a|$. As a common assumption in previous work (e.g. [Pokharel and Das, 2023, Liu et al., 2020, 2021]), all preferences are strict, which means that both the minimal gap of player $\Delta = \min_{j \in \mathcal{N}} \Delta_j$ and the minimal gap of arm $\Delta^a = \min_{k \in \mathcal{K}} \Delta_k^a$ are positive. Moreover, we consider the reasonable case where the difficulty level of arms' preferences learning is comparable with players' up to a positive constant $D \in (0, \infty)$. Specifically, we assume $D\Delta^a \geq \Delta_j$ for all $j \in \mathcal{N}$ in the main part of the paper (except for Section 4). Throughout the time horizon $T$, every player and arm will learn about their own preferences through interactions and want to match with one from the other side with higher utility. We use the notation $j_1 \succ_k j_2$ to indicate that arm $k$ prefers player $j_1$ to player $j_2$ and the similar notation $k_1 \succ_j^a k_2$ to represent that player $j$ prefers arm $k_1$ to arm $k_2$.

---

[1]This assumption guarantees that each player can match with at least one arm. However, by adjusting the exploration phase, our algorithm remains effective even when $N > K$, with regret bounded by $O(N \log T / \Delta^2)$.

At each time step $t \leq T$, each player $j$ pulls an arm $I_j(t)$ simultaneously. If there exists one player pulling the arm $k$, we assume that the arm $k$ will choose to match with the player rather than staying unmatched since all utilities are non-negative. When there are multiple players pulling arm $k$, a conflict arises, and arm $k$ will choose to match one of the candidates based on its strategy (see details in Section 2.2). The unchosen players will get rejected and obtain no reward. Denote the winning player on arm $k$ at time step $t$ by $A_k(t)$. Let $C_j(t)$ represent the rejection indicator of player $j$ at time step $t$. $C_j(t) = 1$ indicates that player $j$ gets rejected and $C_j(t) = 0$ otherwise. When a match succeeds between player $j$ and arm $k$, both player $j$ and arm $k$ receive stochastic rewards sampled from the fixed latent 1-subgaussian distributions with mean $u_{jk}$ and $u_{kj}^a$, respectively. In this paper, we consider the general fully decentralized setting, i.e., no direct communication among players is allowed, and there is no central organizer or extra external information such as observation.

**No Centralized Organizer and Explicit Communication.** When considering learning in matching markets, several studies ([Min et al., 2022, Jagadeesan et al., 2023]) assume the existence of a central platform capable of directly determining the matching outcome. However, real-world applications typically involve participants who act individually and independently, reflecting a decentralized setting where there is no central organizer or explicit communication between players to facilitate direct coordination among players. Additionally, due to privacy considerations, participants may opt out of disclosing their received rewards ([Rees-Jones and Skowronek, 2018]). For scalability reasons, decentralized solutions are also favored ([Larsson, 2018]). Notably, the majority of the related works referenced in our paper consider matching bandits from a decentralized perspective and emphasize the importance of the decentralized setting ([Sankararaman et al., 2021, Liu et al., 2021]).

**No Observation of Winning Players.** In the literature on matching bandits, observation of winning players (which assumes that all players can observe all the winning players on all arms) is a strong but widely used assumption. Even when some arms are never selected by the player, the player can also get their information based on observation. This assumption greatly helps players to learn arms' preferences and other players' actions. Liu et al. [2021] incorporate the observation to design a conflict-avoid algorithm, Kong and Li [2023] use the observation to help players infer others' learning progress easily. However, it will be more challenging but more desirable to throw away the assumption. In real applications, the common case is that a player will only be informed of her own result (acceptance or rejection) rather than being aware of every accepted player. The assumption of no observation also captures the fully decentralized scenario, i.e. players take actions only based on their own matching histories, without access to others' information.

## 2.1 REGRET FOR BOTH SIDES

Before we introduce the definition of regret, we recall the definition of matching stability, which is an important issue when considering matching bandits in matching markets.

A matching between the player side and the arm side is stable if there does not exist a (player, arm) pair such that each one prefers the other partner to the current matched partner. For each player $j$, her optimal stable arm $\overline{m}_j$ is the arm with the highest utility among her matched arms in all possible stable matchings while her pessimal stable arm $\underline{m}_j$ is the matched arm with the lowest utility. For each arm $k$, its optimal stable player $\overline{m}_k^a$ is the player with the highest utility among its matched players in all possible stable matchings while its pessimal stable player $\underline{m}_k^a$ is the matched player with the lowest utility. We define stable regret by comparing the utility of the matched pair with the stable pair. The player-optimal and player-pessimal stable regret for player $j$ are defined as follows, respectively:

$$\overline{R}_j(T) = \mathbb{E}[\sum_{t=1}^{T}(u_{j\overline{m}_j} - (1 - C_j(t))u_{jI_j(t)})],$$

$$\underline{R}_j(T) = \mathbb{E}[\sum_{t=1}^{T}(u_{j\underline{m}_j} - (1 - C_j(t))u_{jI_j(t)})].$$

Similarly, the arm-optimal and arm-pessimal stable regret for arm $k$ are defined as follows, respectively:

$$\overline{R}_k^a(T) = \mathbb{E}[\sum_{t=1}^{T}(u_{k\overline{m}_k^a}^a - u_{kA_k(t)}^a)],$$

$$\underline{R}_k^a(T) = \mathbb{E}[\sum_{t=1}^{T}(u_{k\underline{m}_k^a}^a - u_{kA_k(t)}^a)].$$

Furthermore, the Gale-Shapley (GS) algorithm outlined in [Gale and Shapley, 1962] ensures the existence of a stable matching. Consequently, the aforementioned definition is justified.

**Player-optimal Stable Regret.** The optimal stable regret is defined with respect to the optimal stable pair that has higher utility than the pessimal stable pair. Consequently, achieving sublinear optimal stable regret is considered more challenging and desirable. However, a classical result in [Gale and Shapley, 1962] indicates the impossibility of simultaneously achieving sublinear regret that is both player-optimal and arm-optimal. Gale and Shapley also introduce the Gale-Shapley (GS) algorithm, which secures optimal stable matching for the proposing side. We aim to investigate whether a similar result holds in the context of the two-sided learning setting. Therefore, in this paper, our primary focus lies on player-optimal stable matching.

## 2.2 ARMS' STRATEGIES

In this section, we specify the strategies for arms to choose matched pairs among candidates. Furthermore, rather than focusing on a particular strategy like in [Pagare and Ghosh, 2023], we explore the broader scenario where arms have the flexibility to employ diverse strategies, provided that such strategies remain aligned with the overarching goal of maximizing their individual utilities.

If arms are aware of their preferences beforehand, they can straightforwardly select the option with the highest utility among the candidates to optimize their rewards. However, in the context of a two-sided learning setting, arms lack awareness of their own utilities and must engage in a learning process through interactions. This implies that arm $k$ can only make a selection to match one player based on past rewards received. Nevertheless, with the accumulation of samples from various players, arms' estimations of their own utilities become more accurate. For arms that adopt "rational" strategies, once they have gathered a sufficient number of samples for each player, they are likely to choose to match with the player exhibiting the highest utility with a high probability.

We will formally introduce the rational condition to describe arms' learning strategies below, and we focus on such strategies for arms in the main part of the paper (except for Section 4). We will see later that common bandit learning algorithms satisfy this condition (e.g., UCB, empirical estimator), showing that such an assumption on arms' behavior is not restricted. The empirical mean associated with player $j$ estimated by arm $k$ is denoted by $\hat{u}^a_{kj}$ and the matched times associated with player $j$ estimated by arm $k$ is denoted by $N^a_{kj}$. Define event $\mathcal{E}^a = \{\forall j \in \mathcal{N}, k \in \mathcal{K}, |\hat{u}^a_{kj} - u^a_{kj}| < 2\sqrt{\frac{\log T}{N^a_{kj}}}\}$. The event $\mathcal{E}^a$ represents that the samples' quality is not too bad, i.e., the empirical means are not very far from true values at every time slot. We will show in our proof that $\mathcal{E}^a$ is a high-probability event since all samples are drawn from sub-gaussian distributions.

**Definition 1** (Arm's Rational Condition). *We say arm $k$ adopts a strategy that satisfies $R$ rational condition, if after collecting $R\frac{\log T}{(\Delta^a)^2}$ samples for every player, conditional on $\mathcal{E}^a$, arm $k$ will choose to match with the player with highest utility among the candidates.*

Rational strategies not only guarantee that arms are rationally motivated to maximize their individual utilities but also facilitate the prompt provision of valuable feedback for players. When adopting rational strategies, arms exhibit a tendency to avoid choosing suboptimal candidates extensively, as long as the quality of samples remains reasonably satisfactory. Consequently, players will not receive inaccurate feedback a lot.

As mentioned, the rational condition is not a strict con-straint. Through simple calculation and scaling, we can see that numerous widely employed bandit-learning techniques, including the Upper Confidence Bound (UCB) policy and those following the empirical leader, meet the criteria for the rational condition with $R = 16$. Moreover, our assumption for arms' strategies covers scenarios where some arms use empirical mean estimators to choose which invitation to accept, while other players use UCB estimators to select candidates.

## 3 ROUND-ROBIN ETC ALGORITHM

In this section, we propose our algorithm for players: Round-Robin ETC which obtains an asymptotic $O(\log T)$ player-optimal stable regret. By introducing Round-Robin ETC, we demonstrate that, even in the context of two-sided unknown preferences, it is possible to attain the optimal stable matching for the proposing side with low regret.

### 3.1 CHALLENGES AND SOLUTIONS

In this subsection, we will discuss some unique challenges in the two-sided learning matching bandits, as well as how our proposed techniques address the challenges. Then, we give a brief introduction of the major phases in our algorithm.

A unique challenge brought by the two-sided learning setting lies in the asymmetry of the learning ability on both sides. Intuitively, it will be harder for arms to collect enough samples since players can choose arms proactively while arms can only passively choose one player from the candidates. Despite this asymmetry, it is imperative for arms to expediently and accurately learn their preferences, as this early learning is crucial for ensuring that players receive accurate information during conflicts. In essence, addressing this asymmetry is crucial for the overall success of the learning process in a two-sided learning setting.

Another typical challenge lies in the absence of explicit communication channels or direct observation of conflict results. Reaching the correct optimal stable matching requires cooperation between the independent players and arms given limited communication. It is challenging but crucial to let players decide on when to end their individual exploration and to start a collective matching process. Players face difficulties in discerning the exploration progress of other players, and it becomes even more challenging to understand the exploration progress of arms, given that players can only infer this from the passive actions of arms (i.e., selecting one player from the candidates). Consequently, determining the optimal timing and method to foster cooperation and converge to stable matching poses significant challenges for those independent players.

To tackle these challenges, we initially acknowledge that in conflict-free scenarios, i.e., when only one player pulls one

arm, a successful match occurs, generating a clear sample for both the player and the arm. Leveraging the concept of round-robin exploration, we aim to avoid conflicts and facilitate simultaneous preference learning for both the player-side and the arm-side. Additionally, we integrate confidence bounds to enable players to measure their individual exploration progress and wait for arms to accumulate sufficient samples. In order to estimate the learning progress of other players, we design decentralized communication through deliberate conflicts, which allow players to send and infer information. Specifically, players will deliberately compete for an arm, trying to send information by letting other players get rejected or to receive information by inferring from the rejection indicators. Furthermore, we carefully design the algorithm such that players can enter exploitation as soon as possible, i.e., they do not need to wait until all others have learned their preferences accurately. The intuitive idea is that, if a player is to start exploitation, she only needs to make sure that any other player that could potentially "squeeze" her out has already entered (or is also about to enter) exploitation.

Together with these analyses, we provide a brief introduction of our algorithm. Firstly, the algorithm will assign a distinct index to each player. Next, players will do rounds of round-robin exploration. After every round of exploration, players will communicate their progress of preference learning to decide on whether to start matching. If players decide to start matching, they will run the Gale-Shapley (GS) algorithm and occupy their potential optimal stable arm till the end. Otherwise, the players will start a new round of exploration.

### 3.2 ROUND-ROBIN ETC

The Algorithm 1 consists of 3 phases: "Index Assignment", "Round Robin" and "Exploitation". Players will enter the "Index Assignment" phase and the "Round Robin" phase simultaneously but may leave the "Round-Robin" phase for the "Exploitation" phase at different time steps.

In the "Index Assignment" phase (Line 1), every player will receive a distinct index. To be specific (see procedure *INDEX-ASSIGNMENT*), every player will keep pulling arm 1 until the first time, say step $t$, she doesn't get rejected. She will be assigned index $t$ and then move to pull the next arm, i.e., arm 2. Since there can only be one player that successfully matches with arm 1 at each time step, after $N$ time steps, all players can receive different indices.

In the "Round Robin" phase (Line 3-18), the players will (1) explore the arms without conflict, (2) communicate on their progress of exploration, and (3) start matching or update their indices and available arms in a round based way. Specifically, each round comprises three sub-phases: (1) exploration, (2) communication, and (3) update. A player will leave the "Round Robin" phase when she finds out her optimal stable arm confidently. Then, she will enter the "Ex-

ploitation" phase and occupy her potential optimal stable arm, say arm $k$, making arm $k$ unavailable to other players. Denote the set of players that are still in the "Round Robin" phase by $\mathcal{N}_2$, the number of remaining players by $N_2$, the available set of arms by $\mathcal{K}_2$, and the number of available arms by $K_2$. We further elaborate on the three sub-phases in "Round Robin" below.

---

**Algorithm 1** Round Robin ETC (for a player $j$)

   # Phase 1: Index Assignment
1: Index $\leftarrow$ *INDEX-ASSIGNMENT($N, \mathcal{K}$)*
   # Phase 2: Round Robin
2: $N_2 \leftarrow N, \mathcal{K}_2 \leftarrow \mathcal{K}, K_2 \leftarrow K$  # $N_2$ is the number of remaining players in Phase 2, $\mathcal{K}_2$ is available arms
3: **while** OPT$= \emptyset$ **do**
   # when $j$ hasn't found her potential optimal stable arm
   #Sub-Phase: Exploration
4:    (Success, $\hat{\boldsymbol{u}}_j, \boldsymbol{N}_j$) $\leftarrow$ *EXPLORATION*(Index, $K, K_2, \mathcal{K}_2, \hat{\boldsymbol{u}}_j, \boldsymbol{N}_j$)
   #Sub-Phase: Communication
5:    Success $\leftarrow$ *COMM*(Index, Success, $N_2, K_2, \mathcal{K}_2$)
   #Sub-Phase: Update
6:    OPT $\leftarrow$ *GALE-SHAPLEY*,$N_1 \leftarrow N_2, \mathcal{K}_1 \leftarrow \mathcal{K}_2$
   # $N_1, \mathcal{K}_1$ are temporary parameters to help update $N_2, \mathcal{K}_2$
7:    **if** Success$= 1$ **then Break while**
   #successful players will enter the exploitation phase
8:    **end if**
9:    **for** $t = 1, ..., N_2 K_2$ **do**#check arms' availability
10:      **if** $t=$(Index$-1)K_2+m$ **then**
11:       Pull arm $k$ that is $m$-th arm in $\mathcal{K}_2$
12:       **if** $C_j=1$ **then** $\mathcal{K}_1 \leftarrow \mathcal{K}_1 \backslash \{k\}, N_1 \leftarrow N_1-1$
13:       **end if**
14:      **end if**
15:    **end for**
16:    $N_2 \leftarrow N_1, \mathcal{K}_2 \leftarrow \mathcal{K}_1$ #update available arms and number of players
17:    Index $\leftarrow$ *INDEX-ASSIGNMENT($N_2, \mathcal{K}_2$)*
18: **end while**
   #Phase 3: Exploitation Phase:
19: Pull OPT arm

---

   **procedure** *INDEX-ASSIGNMENT*$(N, \mathcal{K})$
1: $\pi \leftarrow \mathcal{K}[1]$
2: **for** $t = 1, 2, ..., N$ **do**
3:    Pull arm $\pi$
4:    **if** $C_j = 0, \pi = \mathcal{K}[1]$ **then** Index $\leftarrow t, \pi \leftarrow \mathcal{K}[2]$
5:    **end if**
6: **end for**
7: **return** Index

---

1. Exploration (Line 4, see Algorithm 2 *EXPLORATION*). Every player will explore available arms according to the index to avoid conflict, and every exploration will last for $K_2 K^2 \lceil \log T \rceil$ time steps. Based on the distinct index and the assumption that $K \geq N$, each arm is pulled by at most one player at each time step during the exploration,

---

**Algorithm 2** *EXPLORATION* (for player $j$)

**Require:** Index, $K_1, K, \mathcal{K}, \hat{\boldsymbol{u}}_j, \boldsymbol{N}_j$
 1: **for** $t = 1, 2, ..., K K_1^2 \lceil \log T \rceil$ **do**
 2:     Pull (Index $+ t$)mod $K = m$-th arm in $\mathcal{K}$ and update $\hat{u}_{jk}, N_{jk}$
 3: **end for**
 4: **if** $\forall k_1 \neq k_2 \in \mathcal{K}$, $\text{UCB}_{jk_1} < \text{LCB}_{jk_2}$ or $\text{LCB}_{jk_1} > \text{UCB}_{jk_2}$ **then** Success $\leftarrow 1$
     # whether the player achieves a confident estimation
 5: **end if**
 6: **return** Success, $\hat{\boldsymbol{u}}_j, \boldsymbol{N}_j$

---

**Algorithm 3** *COMM* (for player $j$)

**Require:** Index, Success, $N, K, \mathcal{K}$
 1: **for** $i = 1, 2, ..., N$,t_index $= 1, 2, ..., N$, r_index $= 1, 2, ..., N$, r_index$\neq$t_index **do**
     # player with t_index is transmitter, r_index is receiver
 2:     **for** $m = 1, 2, ..., K$ **do**
     # communication process is through conflicts on $m$-th arm
 3:       **if** Index=t_index # if transmitter **then**
 4:        Pull the $m$-th arm in $\mathcal{K}$ if Success$= 0$
 5:       **end if**
 6:       **if** Index=r_index # if receiver **then**
 7:        Pull $m$-th arm in $\mathcal{K}$, Success$\leftarrow 0$ if $C_j = 1$
 8:       **end if**
 9:     **end for**
10: **end for**
11: **return** Success

---

preventing any conflicts. Player $j$ will update her empirical mean $\hat{\boldsymbol{u}}_j$ and the matched times $\boldsymbol{N}_j$ throughout the exploration. To measure her progress in preference learning, player $j$ will incorporate confidence bounds. The notions of upper confidence bound "UCB" and lower confidence bound "LCB" are defined as follows:

$$\text{UCB}_{jk} = \hat{u}_{jk} + c\sqrt{\frac{\log T}{N_{jk}}}, \text{LCB}_{jk} = \hat{u}_{jk} - c\sqrt{\frac{\log T}{N_{jk}}},$$

where $\hat{u}_{jk}$ denotes the empirical mean and $N_{jk}$ denotes the times player $j$ is matched with arm $k$. Let $c = \max\{2, \frac{\sqrt{R}D}{2} + 1\}$. We say that when a player $j$ achieves a confident estimation on the arm set $\mathcal{K}^*$ if for every $k_1, k_2 \in \mathcal{K}^*$ such that $k_1 \neq k_2$, either $\text{UCB}_{jk_1} < \text{LCB}_{jk_2}$ or $\text{LCB}_{jk_1} > \text{UCB}_{jk_2}$ holds.

2. Communication (Line 5, see Algorithm 3 *COMM*). The players will communicate through deliberate conflicts in an index-based order. This sub-phase lets players communicate on their progress of exploration. Specifically, they will communicate whether they have achieved confident estimations and the communication proceeds pairwise following the order of the index. The player with index 1 will first serve as a transmitter, sending information to the player with index 2, then to players with index 3,

4 and so on. After the player with index 1 has finished sending information to others, the player with index 2 will be the transmitter, then the player with index 3 and so on. The player who wants to receive information is the receiver.

The communication subphase conducts all pairwise communication between all pairs of remaining players on all available arms for $N_2$ times. Specifically, for every pair of different remaining players $j_1$ and $j_2$, communication occurs on every available arm for $N_2$ times. Every communication is conducted through a deliberate conflict on a communication arm of $\mathcal{K}_2$ between a transmitter and a receiver. The player with index "t_index", denoted as player $j_1$, serves as the transmitter, and the player with index "r_index" is the receiver (Line 1 in Algorithm 3). Suppose $j_2$ is the receiver, and arm $k$ is the communication arm, i.e. the $m$-th arm of $\mathcal{K}_2$ (Line 2 in Algorithm 3). The receiver $j_2$ will choose arm $k$ to receive information. The transmitter $j_1$ will choose arm $k$ only when she fails to achieve a confident estimation or has been rejected when receiving others' information in the previous time steps during the communication sub-phase. Other players will pull an arbitrary arm $k' \neq k$.

If a player achieves a confident estimation and never gets rejected when receiving others' information during the communication sub-phase, we say that the player obtains successful learning. Note that if a player obtains successful learning, it means that with high probability, the remaining players that may "squeeze" her out on the available arms all achieve confident estimations (and all obtain successful learning). We use "Success" in the pseudocode (Line 4, 5, 7) to denote the success signal, and "Success$= 1$" indicates that the player obtains successful learning while "Success$= 0$" otherwise. We call the players who obtain successful learning successful players, and others are called unsuccessful players.

3. Update (Line 6-17). The successful players will be able to find out their potential optimal stable arms, and unsuccessful players will update their indices, the number of remaining players $N_2$, and the set of available arms $\mathcal{K}_2$. The first procedure *GALE-SHAPLEY* ([Gale and Shapley, 1962]) enables successful players to match their potential optimal stable arms. Then successful players will enter the "Exploitation" phase, and unsuccessful players will update the available arms in order. Specifically, when $t = (n - 1)K_2 + m$ in Line 10, the player with index $n$, suppose player $j$, will pull the $m$-th arm in $\mathcal{K}_2$, suppose arm $k$, to check its availability. If player $j$ gets rejected, then she will kick arm $k$ out of the available arm set. Lastly, unsuccessful players will update their indices by the *INDEX-ASSIGNMENT* function and start a new round.

In the "Exploitation" phase (Line 19), every player will keep pulling her potential optimal stable arm till the end.

**Rationale of the Communication.** Note that in the scenario where only a subset of players completes their preference learning and initiates the exploitation of stable pairs within this subset, the stable pairs obtained may not align with those in the original stable matching. Previous research often relies on specific preference frameworks to address this issue, such as uniqueness consistency and global ranking assumptions. Nevertheless, when dealing with a general preference structure, it is common for other players with higher priority to potentially squeeze those settled players out from their current pairs, resulting in chaos in the matching markets. Communication serves the purpose of conveying information about whether these high-priority players have finished their explorations. Therefore, the communication process indeed helps the players match with their optimal stable arms effectively and with minimal cost.

**Example 1** (Example of Round Robin phase). *Consider a matching market with three arms denoted as $a_1, a_2, a_3$ and three players denoted as $p_1, p_2, p_3$. The individual preferences are outlined as follows:*

$$p_1 : a_1 \succ a_2 \succ a_3, \quad a_1 : p_1 \succ^a p_3 \succ^a p_2,$$
$$p_2 : a_3 \succ a_2 \succ a_1, \quad a_2 : p_1 \succ^a p_3 \succ^a p_2,$$
$$p_3 : a_2 \succ a_3 \succ a_1, \quad a_3 : p_3 \succ^a p_1 \succ^a p_2.$$

*If arms consistently provide accurate feedback once any player achieves a confident estimation (an event with high probability), the Round Robin phase may proceed as follows:*

Table 2: An example of Round Robin Phase.

| round | remaining players | confident players | successful players | available arms |
|---|---|---|---|---|
| 1 | $\{p_1, p_2, p_3\}$ | $\{p_3\}$ | $\emptyset$ | $\{a_1, a_2, a_3\}$ |
| 2 | $\{p_1, p_2, p_3\}$ | $\{p_1, p_3\}$ | $\{p_1, p_3\}$ | $\{a_1, a_2, a_3\}$ |
| 3 | $\{p_2\}$ | $\{p_2\}$ | $\{p_2\}$ | $\{a_3\}$ |

*The round-robin phase comprises three rounds, resulting in two rounds of communication.*

*In the first round, only player $p_3$ attains a confident estimation. Consequently, during communication, player $p_1$ will initially pull the three arms in order twice, serving as the transmitter. During the first three pulls, player $p_2$ will sequentially pull the three arms to receive information, serving as a receiver. Subsequently, during the next three pulls, player $p_3$ will serve as the receiver, i.e. pull the three arms to obtain information. However, when player $p_3$ pulls arms 1 and 2, she will get rejected, as these arms prefer player $p_1$. Consequently, no player achieves successful learning in this round.*

*In the second round, players $p_1$ and $p_3$ achieve confident estimations. During this round of communication, since players $p_1$ and $p_3$ have achieved confident estimations, they will not pull the same arm as the receiver when transmitting information. Consequently, receivers will not get rejected when receiving their information. Player $p_2$ will pull the*

*same arm as the receiver. However, since players $p_1$ and $p_3$ are preferred over player $p_2$ on all arms, they will also not get rejected when receiving information from $p_2$. This implies that players $p_1$ and $p_3$ obtain successful learning and will proceed to the exploitation phase.*

*In the last round, with only one player remaining, there is no communication. Having achieved a confident estimation, player $p_2$ will leave the round-robin phase for the exploitation phase after this round.*

### 3.3 REGRET ANALYSIS

**Theorem 1.** *If every player runs Algorithm 1, and arms adopt $R$ rational strategies, then the optimal stable regret of any player $j$ can be upper bounded by:*

$$\overline{R}_j(T) \leq N + K^3 r \lceil \log T \rceil + Nr(KN(N-1) + N + K + 1) + 4KN + 2$$
$$= O(\frac{K \log T}{\Delta^2}).$$

*Moreover, the arm-pessimal stable regret for any arm $k$ can also be upper bounded by:*

$$\underline{R}_k^a(T) \leq N + K^3 r \lceil \log T \rceil + Nr(KN(N-1) + N + K + 1) + 4KN + 2$$
$$= O(\frac{K \log T}{\Delta^2}),$$

*where $r$ equals to $\lceil \frac{4(c+2)^2}{K^2 \Delta^2} \rceil$ and $c = \max\{2, \frac{\sqrt{R}D}{2} + 1\}$.*

*Proof Sketch.* We provide only a proof sketch for the player-optimal stable regret, with similar analysis applicable to derive the result for the arm regret. The complete proof is available in Appendix B.

Define the event $\mathcal{E} = \{\forall j \in \mathcal{N}, k \in \mathcal{K}, |\hat{u}_{jk} - u_{jk}| < 2\sqrt{\frac{\log T}{N_{jk}}}\}$. We can decompose the regret depending on whether $\mathcal{E}$ and $\mathcal{E}^a$ holds, i.e.

$$\overline{R}_j(t) \leq \mathbb{E}[\sum_{t=1}^{T}(u_{j\overline{m}_j} - (1 - C_j(t))u_{jI_j(t)})|\mathcal{E} \cap \mathcal{E}^a]$$
$$+ TPr[\neg\mathcal{E}] + TPr[\neg\mathcal{E}^a].$$

While the probability of $\neg\mathcal{E}$ and $\neg\mathcal{E}^a$ can be upper bounded by a $\frac{1}{T}$ factor, we only need to bound the regret conditional on $\mathcal{E} \cap \mathcal{E}^a$. By the design of the algorithm, we can easily find out that the initialization phase lasts for $N$ time steps, which means there will be at most $N$ regret caused by the initialization phase. As for the other two phases, we can prove the following statements:

- Conditional on $\mathcal{E}$ and $\mathcal{E}^a$, with probability more than $1 - \frac{2}{T}$, when a player achieves a confident estimation on the available arm set $\mathcal{K}_2$, the arms in $\mathcal{K}_2$ give accurate feedback.

- Conditional on $\mathcal{E}$ and $\mathcal{E}^a$, all players can achieve confident estimations after collecting $O(\log T)$ samples in the exploration.

- If arms in $\mathcal{K}_2$ give accurate feedback, conditional on $\mathcal{E}$ and $\mathcal{E}^a$, the successful players will pull their optimal stable arms in the exploitation phase.

Then according to the design of the algorithm and these statements, we can also prove that conditional on $\mathcal{E} \cap \mathcal{E}^a$, after no more than $O(\log T)$ time steps in the round-robin phase, all players will enter the exploitation phase with their correct optimal stable arm with high probability. Combining these all together, we can obtain the results. $\square$

# 4 EXTENDING TO COLLABORATIVE ARMS

In this section, we examine scenarios in which arms implement more complex policies. We demonstrate that through collaboration between both parties, the assumption of learning difficulty can be eliminated. Furthermore, with support from the arms, players can achieve low regret. Specifically, we analyze the collaborative case with arbitrary learning difficulties and more complex arm strategies beyond rational strategies, in contrast to the previous scenario that assumed $\Delta^a > D\Delta$ and rational strategies for arms. The primary objective of this section is to serve as a preliminary exploration, laying the groundwork for a more comprehensive investigation into various arm strategies. Our aim is to understand how these diverse strategies employed by the arms can influence outcomes within the matching market.

**High-Level Idea.** The necessity for assuming learning difficulty arises from our objective to guarantee that, when a player possesses a confident estimation, the arms learn their own preferences accurately and offer precise feedback, thereby ensuring effective communication. However, if arms are granted the ability to employ more complex strategies, such as employing forced rejection to indicate whether she has completed preference learning, we can effectively address the previously mentioned issue.

## 4.1 PLAYERS' STRATEGIES

Since the new algorithm (Algorithm 4) is similar to Algorithm 1, we will provide a brief overview, focusing primarily on the differences.

Players are initially assigned distinct indices. They then alternate between communication and exploration until every participant, including the arms, obtains a confident estimation. Specifically, players communicate through deliberate conflicts, gaining insights into others' learning processes. If there is a participant who hasn't achieved a confident estimation, all players return to exploration. Once all participants have confident estimations, players execute the *GALE-SHAPLEY* algorithm to identify potential optimal stable arms and occupy them until the end.

---

**Algorithm 4** Round Robin ETC with help from arms (for a player $j$)

---

1: Index $\leftarrow$ *INDEX-ASSIGNMENT(N, $\mathcal{K}$)*
2: **while** OPT$= \emptyset$ **do**
   \# when $j$ hasn't found her potential optimal stable arm yet
3:     (Success, $\hat{\boldsymbol{u}}_j$, $\boldsymbol{N}_j$) $\leftarrow$ *EXPLORATION(Index, K, K, $\mathcal{K}$, $\hat{\boldsymbol{u}}_j$, $\boldsymbol{N}_j$)*
4:     Success $\leftarrow$ *COMM_ARM(Index, Success, $N_2$, $K_2$, $\mathcal{K}_2$)*
5:     **if** Success$= 1$ **then**
6:         OPT $\leftarrow$ *GALE-SHAPLEY*
7:     **end if**
8: **end while**
9: Pull OPT arm

---

**Algorithm 5** *COMM_ARM* (for player $j$)

---

**Require:** Index, Success, $N, K, \mathcal{K}$
1: **if** Success$= 1$ **then** Pull arm 1
2: **else**    Pull arm 2
3: **end if**
4: **for** $k = 1, 2, ..., K, t = 1, 2, ..., N$ **do** Pull arm $k$
5:     **if** $t =$ Index, $C_j = 1$ **then** Success$\leftarrow 0$
6:     **end if**
7: **end for**
8: **return** Success

---

Recall the notions of upper confidence bound "UCB" and lower confidence bound "LCB" for player $j$:

$$\text{UCB}_{jk} = \hat{u}_{jk} + c\sqrt{\frac{\log T}{N_{jk}}}, \text{LCB}_{jk} = \hat{u}_{jk} - c\sqrt{\frac{\log T}{N_{jk}}}, \quad (1)$$

where $\hat{u}_{jk}$ denotes the empirical mean and $N_{jk}$ denotes the times player $j$ is matched with arm $k$. In this section, let $c = 2$. We say that when a player $j$ achieves a confident estimation if for every $k_1, k_2 \in \mathcal{K}$ such that $k_1 \neq k_2$, either $\text{UCB}_{jk_1} < \text{LCB}_{jk_2}$ or $\text{LCB}_{jk_1} > \text{UCB}_{jk_2}$ holds.

Regarding communication (Line 4, see Algorithm 5 *COMM_ARM*), players will initially report their learning progress to arm 1 and subsequently receive feedback from arm 1, then arm 2, and so on. Specifically, players with confident estimations will pull arm 1, while others will pull a different arm. Subsequently, players will take turns receiving information about others' learning progress. More precisely, each player will pull each arm for $N$ times, and successful learning is achieved only if she wins at the time step corresponding to her index. Successful learning implies that every participant has a confident estimation. Thus, if a player achieves successful learning, she will execute the *GALE-SHAPLEY* algorithm to determine her potential optimal stable arm.

## 4.2 ARMS' STRATEGIES

Arms continuously update their empirical means and matched times throughout the entire time horizon $T$ and as-

sess their learning progress using confidence bounds. Additionally, arms act as communication intermediaries. Specifically, arms convey information to players by intentionally rejecting certain candidates.

Briefly speaking, arms will alternate between communication and selection (in Algorithm 6). The communication periods for arms coincide with those for players. While not engaged in communication, the arms will choose players myopically, selecting the most preferred candidates based on empirical means.

---

**Algorithm 6** Arm Strategy (for an arm $k$)

---

1. Convey information during the communication period.
2. When not in communication, choose the most preferred candidate based on empirical means, and keep updating estimations.

---

Similarly, define the notions of upper confidence bound "UCB" and lower confidence bound "LCB" for arm $k$:

$$\text{UCB}_{kj}^a = \hat{u}_{kj}^a + c\sqrt{\frac{\log T}{N_{kj}^a}}, \text{LCB}_{kj}^a = \hat{u}_{kj}^a - c\sqrt{\frac{\log T}{N_{kj}^a}}, \quad (2)$$

where $\hat{u}_{kj}^a$ denotes the empirical mean and $N_{kj}^a$ denotes the times arm $k$ is matched with player $j$. Let $c = 2$. We say that when an arm $k$ achieves a confident estimation if for every two player $j_1, j_2$ such that $j_1 \neq j_2$, either $\text{UCB}_{kj_1}^a < \text{LCB}_{kj_2}^a$ or $\text{LCB}_{kj_1}^a > \text{UCB}_{kj_2}^a$ holds.

---

**Algorithm 7** *COMM_ARM* (for arm $k^*$)

---

1: record the number of candidates as $N_p$ if arm 1
2: **for** $k = 1, 2, ..., K, t = 1, 2, ..N,$ **do**
3:     **if** arm 1 **then** if achieves a confident estimation and $N_p = N$, accept the player with Index $t$
4:     **else**   if achieves a confident estimation, accept the player with Index $t$
5:     **end if**
6: **end for**

---

Regarding communication (See Algorithm 7 *COMM_ARM*), arm 1 first checks if the number of invitations equals the number of players and then selects an arbitrary candidate. Over the next $KN$ time steps, each arm communicates information about whether it has achieved a confident estimation to the players, and arm 1 additionally conveys information about whether all players have achieved confident estimations. Specifically, during time step $t$ in the first period of $N$ time steps, if arm 1 has a confident estimation and receives $N$ invitations during the previous check, it selects the candidate with index $t$. Similarly, for arm $k$, it chooses the candidate with index $t$ during time step $t$ in the $k$-th period of $N$ time steps only if it has a confident estimation. After the communication phase, if each participant attains a confident estimation, every player will be accepted

at the designated time based on her index for each arm. Subsequently, the players are about to initiate a collective matching process.

Note that in our algorithms, we consider that all arms have knowledge of the indices of all players. This can be easily adjusted by incorporating an index assignment procedure on each arm, which only requires $KN$ time steps in total.

### 4.3 REGRET ANALYSIS

The following theorem demonstrates the effectiveness of our algorithms. The detailed proof can be found in the Appendix D.

**Theorem 2.** *If all players run Algorithm 4 and arms adopt strategies Algorithm 6, then the optimal stable regret of any player $j$ can be upper bounded by* [2] :

$$\overline{R}_j(T) \leq N + K^3 r \lceil \log T \rceil + r(1 + KN) + 4KN$$
$$= O(\frac{K \log T}{\Delta_*^2}),$$

*where $r$ equals to $\lceil \frac{64}{K^2 \Delta_*^2} \rceil$ and $\Delta_* = \min\{\Delta, \Delta^a\}$.*

## 5 CONCLUSION

Inspired by the classical GS algorithm, in this work, we study the convergence to optimal stable matching for the proposing side in the two-sided learning matching markets. Throwing away many previous assumptions such as observations and special preference structures in matching bandits literature, we study the more general case and consider strategies for both sides. We model the passive side, namely the arm side, with a reasonable "Rational Condition", where their objective is to maximize their individual rewards. Then, on the proactive side, i.e., the player side, we introduce the Round-Robin ETC algorithm, incorporating various techniques to tackle challenges arising from unreliable feedback from arms and the absence of information and communication. Through rigorous analysis, we demonstrate that the optimal matching for the proposing side can be achieved with high probability. Moreover, our algorithm achieves an $O(\log T)$ player-optimal stable regret, which matches the order of the state-of-the-art guarantee in the simpler one-sided learning setting. The simulations provided in Appendix E further validate our results. To summarize, our work contributes to the understanding of the convergence dynamics in two-sided learning matching markets under the described conditions. Subsequent research directions may involve examining cases where arms adopt other more strategic and sophisticated policies. Furthermore, exploring the dynamics of the strategic interactions between the player-side and the arm-side could serve as an intriguing avenue for further study.

---

[2]Similar result for arm pessimal stable regret can be simply obtained.

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

# A  *GALE-SHAPLEY* IN COMPETING BANDITS

---

**Algorithm 8** *GALE-SHAPLEY* (for a player $j$)

---

**Require:** Success, $N, \mathcal{K}, \hat{\boldsymbol{u}}_j, \boldsymbol{N}_j$
  $i \leftarrow 1$, sort $k \in \mathcal{K}$, let $k_h$ be the arm with $h$-th highest empirical mean in $\mathcal{K}$
  **for** $t = 1, 2, ..., N^2$ **do**
    Pull arm $k_i$
    **if** $C_j = 1$ **then**
      $i \leftarrow i + 1$
    **end if**
  **end for**
  **if** Success$= 1$ **then return** $k_i$ **else return** $\emptyset$
  **end if**

---

In the *GALE-SHAPLEY* algorithm, all the players will propose to their most preferred arms that they haven't encountered rejection on yet.

**Lemma 1.** *[Gale and Shapley, 1962] Suppose player $j$ obtains successful learning. If every player sorts all arms accurately, and every arm gives accurate feedback, then the output of the GALE-SHAPLEY will equal to player $j$'s optimal stable arm.*

Note that if all the left players $\mathcal{N} \setminus \mathcal{N}_2$ occupy their optimal stable arms, all the remaining players can also find out their optimal stable arms through the *GALE-SHAPLEY* algorithm.

# B  REGRET PROOF

Before we analyze the regret bound, we clarify some notations and introduce some lemmas.

Note that $\hat{u}_{jk}$ represents the empirical mean associated with arm $k$, as estimated by player $j$, with $N_{jk}$ indicating the number of times they have been matched. Similarly, $\hat{u}_{kj}^a$ and $N_{kj}^a$ are utilized to denote the empirical mean and matched times associated with player $j$, as estimated by arm $k$. It is important to highlight that players only update empirical means and matched times during the exploration period, whereas arms continuously update their empirical means and matched times throughout the entire time horizon $T$.

Recall that $\mathcal{N}$ denotes the set of players within the entire market, while $\mathcal{N}_2$ signifies the subset of remaining players within the "Round-Robin Phase". Similarly, $\mathcal{K}$ represents the set of arms within the entire market, while $\mathcal{K}_2$ denotes the available arms during the "Round-Robin" phase. The utility gap for player $j$ is denoted as $\Delta_j = \min_{k_1, k_2 \in \mathcal{K}, k_1 \neq k_2} |u_{jk_1} - u_{jk_2}|$, and the utility gap for arm $k$ is denoted as $\Delta_k^a = \min_{j_1, j_2 \in \mathcal{N}, j_1 \neq j_2} |u_{kj_1}^a - u_{kj_2}^a|$. The minimal gap of players is defined as $\Delta = \min_{j \in \mathcal{N}} \Delta_j$, and the minimal gap of arms is defined as $\Delta^a = \min_{k \in \mathcal{K}} \Delta_k^a$. Furthermore, $D$ represents a comparative ratio between two sides, ensuring that $D\Delta^a \geq \Delta_j$ for any $j$ in $\mathcal{N}$.

**Lemma 2.** *(Corollary 5.1 in [Lattimore and Szepesvári, 2020]) Assume that $X_i - u$ are independent, $\sigma$-subgaussian random variables. Then for any $\epsilon \geq 0$,*

$$\Pr[\hat{u} \geq u + \epsilon] \leq \exp(-\frac{n\epsilon^2}{2\sigma^2}) \text{ and } \Pr[\hat{u} \leq u - \epsilon] \leq \exp(-\frac{n\epsilon^2}{2\sigma^2}),$$

*where $\hat{u} = \frac{X_1 + .. + X_n}{n}$.*

**Lemma 3.** *Define the event:* $\mathcal{E} = \{\forall j \in \mathcal{N}, k \in \mathcal{K}, |\hat{u}_{jk} - u_{jk}| < 2\sqrt{\frac{\log T}{N_{jk}}}\}$, *and recall that* $\mathcal{E}^a = \{\forall j \in \mathcal{N}, k \in \mathcal{K}, |\hat{u}_{kj}^a - u_{kj}^a| < 2\sqrt{\frac{\log T}{N_{kj}^a}}\}$, $\Pr[\neg \mathcal{E}] \leq \frac{2KN}{T}$ *and* $\Pr[\neg \mathcal{E}^a] \leq \frac{2KN}{T}$ *hold.*

*Proof.* We can directly get the lemma according to Lemma 2. $\qquad\square$

**Lemma 4.** *Conditional on $\mathcal{E}$ and $\mathcal{E}^a$, with probability more than $1 - \frac{2}{T}$, when a player achieves a confident estimation on the available arm set $\mathcal{K}_2$, the arms in $\mathcal{K}_2$ give accurate feedback.*

Lemma 4 shows that as long as players have confidence on the estimations of arm utilities, the arms will give precise feedback with high probability.

*Proof.* Suppose player $j$ is the first player who achieves a confident estimation, from the design of the algorithm, the remaining arm set $\mathcal{K}_2$ equals the whole arm set $\mathcal{K}$. Suppose arms $k_1, k_2 \in \mathcal{K}$ satisfy $u_{jk_1} - u_{jk_2} = \Delta_j$. Since player $j$ achieves a confident estimation, thus $\text{LCB}_{jk_1} > \text{UCB}_{jk_2}$ conditional on $\mathcal{E}$. During the exploration, all the available arms are explored evenly and without conflict. Note that for player $j$ the rewards received are independent 1-subgaussian random variables, denote the rewards received after being matched with arm $k_1$ during the exploration by $X_1, X_2, ..., X_n$ and the rewards associated with arm $k_2$ by $Y_1, Y_2, ..., Y_n$, where $n = N_{jk_1} = N_{jk_2}$, $Z_1 = X_1 - Y_1, Z_2 = X_2 - Y_2, ..., Z_n = X_n - Y_n$ are independent $\sqrt{2}$-subgaussian random variables. By applying Lemma 2, with probability more than $1 - \frac{2}{T}$ we obtain that:

$$\Delta_j > \frac{Z_1 + ... + Z_n}{n} - 2\sqrt{\frac{\log T}{n}} \geq \text{LCB}_{jk_1} - \text{UCB}_{jk_2} + \sqrt{R}D\sqrt{\frac{\log T}{n}} > D\sqrt{\frac{R\log T}{n}}.$$

Note again that all the available arms are explored evenly and without conflict in the exploration. Thus the matched times for arms satisfy that $N_{kj'}^a \geq n \geq \frac{RD^2 \log T}{\Delta_j^2} \geq \frac{R\log T}{(\Delta^a)^2}$ for every $k \in \mathcal{K}$ and every $j' \in \mathcal{N}$. According to the definition of $R$-rational condition, conditional on $\mathcal{E}^a$, arms give accurate feedback. $\square$

**Lemma 5.** *Conditional on $\mathcal{E}$ and $\mathcal{E}^a$, a player $j$ will achieve a confident estimation on $\mathcal{K}_2$ after no more than $\lceil \frac{4(c+2)^2}{K^2\Delta^2} \rceil$ rounds in the "Round Robin" phase.*

*Proof.* Note that after $\lceil \frac{4(c+2)^2}{K^2\Delta^2} \rceil$ rounds in the "Round Robin" phase, every available arm is matched with player $j$ for at least $\frac{4(c+2)^2 \log T}{\Delta^2}$ time steps during the exploration. Since players only update empirical mean and matched times in the exploration, the matched times of available arms are the same. For $k_1, k_2 \in \mathcal{K}_2$ that $u_{jk_1} > u_{jk_2}$, conditional on $\mathcal{E}$, we have that:

$$\begin{aligned}
\text{LCB}_{jk_1} &= \hat{u}_{jk_1} - c\sqrt{\frac{\log T}{N_{jk_1}}} > u_{jk_1} - (c+2)\sqrt{\frac{\log T}{N_{jk_1}}} \geq \Delta_j + u_{jk_2} - (c+2)\sqrt{\frac{\log T}{N_{jk_2}}} \\
&> \Delta_j + \hat{u}_{jk_2} - (c+4)\sqrt{\frac{\log T}{N_{jk_2}}} > \Delta_j + \hat{u}_{jk_2} + c\sqrt{\frac{\log T}{N_{jk_2}}} - (2c+4)\sqrt{\frac{\log T}{N_{jk_2}}} \\
&= \text{UCB}_{jk_2} + \Delta_j - (2c+4)\sqrt{\frac{\log T}{N_{jk_2}}}.
\end{aligned}$$

We can conclude the lemma based on the fact that $N_{jk_2} \geq \frac{4(c+2)^2 \log T}{\Delta^2}$ after $\lceil \frac{4(c+2)^2}{K^2\Delta^2} \rceil$ round in the "Round Robin" phase. $\square$

We say a player $j'$ can influence player $j$ if there exist a distinct sequence of remaining players $j_0 = j', j_1, ..., j_n = j$ and a sequence of available arms $k_1, ..., k_n$, such that $j_{i-1} \succ_{k_i} j_i$ for $i = 1, 2, ..., n$. Otherwise, we say player $j'$ cannot influence player $j$. The following Lemma indicates the transitivity of influence relation.

**Lemma 6.** *If a player $j_0$ can influence player $j'$, and player $j'$ can influence $j$, then $j_0$ can also influence player $j$.*

*Proof.* Since $j_0$ can influence the optimal stable arm of player $j'$, and player $j'$ can influence the optimal stable arm of player $j$, from the definition, there exist remaining players $j_0, j_1, j_2, ..., j_m = j', ..., j_n = j$ and available arms $k_1, ..., k_n$ that satisfy $j_{i-1} \succ_{k_i} j_i$ for $i = 1, 2, ..., n$ (by emerging two sequences). Note that if one of the following cases happens: (1) there exists $m_1 < m$ that $j = j_{m_1}$, (2) there exists $m_2 > m$ that $j_0 = j_{m_2}$, or (3) there exist no $m_1 < m$ and $m_2 \geq m$ that $j_{m_1} = j_{m_2}$, we can simply conclude the lemma. Otherwise, suppose for $m_1 < m$ and $m_2 \geq m$ that $j_{m_1} = j_{m_2}$ holds, we can find out that the remaining players $j_0' = j_0, ..., j_{m_1-1}' = j_{m_1-1}, j_{m_1}' = j_{m_2}, j_{m_1+1}' = j_{m_2+1}, ..., j_{n'}' = j_n$ and available arms $k_1' = k_1, ..., k_{m_1}' = k_{m_1}, k_{m_1+1}' = k_{m_2+1}, ..., k_{n'}' = k_n$ satisfy $j_{i-1}' \succ_{k_i'} j_i'$ for $i = 1, 2, ..., n'$. Repeat the above process, we can find a distinct sequence of remaining players $j_0 = j_0^*, ..., j_{n^*}^* = j$ and a sequence of available arms $k_1^*, ..., k_{n^*}^*$ that satisfy $j_{i-1}^* \succ_{k_i^*} j_i^*$ for $i = 1, 2, ..., n^*$ which finishes the proof. $\square$

**Lemma 7.** *During a communication, conditional on that arms give accurate feedback, if a player $j$ never gets rejected when receiving, then for any $j' \neq j \in \mathcal{N}_2$, one of the following statements holds:*

*1) player $j'$ achieves a confident estimation on the available arm set $\mathcal{K}_2$,*

*2) player $j'$ cannot influence player $j$.*

*Proof.* We prove the lemma by contradiction. Suppose there exists a player $j' \neq j$ who doesn't achieve a confident estimation on the available arm set $\mathcal{K}_2$ and player $j'$ can influence player $j$. Then there exists a distinct sequence of remaining players $j_0 = j', j_1, ..., j_n = j$ and a sequence of available arms $k_1, ..., k_n$, such that $j_{i-1} \succ_{k_i} j_i$ for $i = 1, 2, ..., n$. Since player $j'$ fails to achieve a confident estimation and arms give precise feedback, there exists $t_1 \leq N_1 K_1$ in the communication process when $j_1$ will get rejected. Similarly, we can conclude that there exists $t_i$ for $i = 1, ..., n$ that $t_i \leq i N_1 K_1$ and at time step $t_i$, player $j_i$ will get rejected. Note that there are at most $N_2$ players remaining, player $j$ will get rejected during the communication which is a contradiction. $\square$

According to the design of Algorithm 1, different players may match their potential optimal stable arms after different rounds in the "Round Robin" phase. GALE-SHAPLEY in [Gale and Shapley, 1962] is used to help players find their potential optimal stable arms. Note that if player $j'$ cannot influence player $j$, the pulls of player $j'$ will not influence the output of the potential optimal arm (i.e. *OPT* in Line 6) for player $j$.

**Lemma 8.** *Conditional on $\mathcal{E}$ and $\mathcal{E}^a$, with probability $1 - \frac{2}{T}$, when a player $j$ obtains successful learning, her potential optimal stable arm equals to her optimal stable arm.*

*Proof.* Note that different players may obtain successful learning after different rounds in the "Round Robin" phase and there may be multiple players obtain successful learning at the same round. We denote the $n$-th (in the round order) set of players to obtain successful learning by $\mathcal{S}(n)$. Define the event: $\mathcal{E}^* = \{$all successful players have correct estimations on $\mathcal{K}_2\} \cap \{$all arms give accurate feedback after a player achieves a confident estimation$\}$. We prove the statement "conditional on $\mathcal{E}^*$, when a player $j$ obtains successful learning, her potential optimal stable arm equals to her optimal stable arm" by mathematical induction. If the above statement holds for players in $\mathcal{S} = \cup_{i=1}^{n-1} \mathcal{S}(i)$, we prove the correctness of the statement for players in $\mathcal{S}(n)$. Note that conditional on $\mathcal{E}^*$, all the players in $\mathcal{S}$ will occupy their optimal stable arms. We now verify that any player $j'$ in $\mathcal{S}(m)$ (where $m = n + 1, ...$) can never influence the optimal stable arm for player $j$ in $\mathcal{S}(n)$. By contradiction, if player $j'$ can influence the optimal stable arm of $j$. Since player $j'$ fails to obtain successful learning at round $n$, $j'$ either fails to achieve a confident estimation or has got rejected when receiving during the $n$-th round's communication. By combing Lemma 7 and Lemma 6, there must exist a player $j_0$ (may equal to player $j'$) who fails to achieve confident estimations and $j_0$ can influence the optimal stable arm of $j$. Then player $j$ must have got rejected when receiving which contradicts the definition of obtaining successful learning. Combining all the above analyses, we can prove the correctness of the statement. Now, based on Lemma 4, we only need to prove the correctness of every player's estimation on the available arm set $\mathcal{K}_2$ conditional on $\mathcal{E}$. Conditional on $\mathcal{E}$, for any $k_1, k_2 \in \mathcal{K}_2$ that $\text{LCB}_{jk_1} > \text{UCB}_{jk_2}$, have:

$$u_{jk_1} > \text{LCB}_{jk_1} > \text{UCB}_{jk_2} > u_{jk_2}.$$

Thus, the correctness of player $j$'s estimation is proved, and the origin statement holds. $\square$

*Proof of Theorem 1.* Let $r = \lceil \frac{4(c+2)^2}{K^2 \Delta^2} \rceil$. By decomposing the player optimal stable regret and using the above lemmas, we obtain that:

$$\overline{R}_j(T) = \mathbb{E}[R_1 + R_2 + R_3 | \mathcal{E} \cup \mathcal{E}^a] + T \Pr[\neg \mathcal{E}] + T \Pr[\neg \mathcal{E}^a] \tag{3}$$
$$\leq N + \mathbb{E}[R_2 + R_3 | \mathcal{E} \cup \mathcal{E}^a] + 4KN \tag{4}$$
$$\leq N + K^3 r \lceil \log T \rceil + r(KN^2(N-1) + N^2 + NK + N) + 4KN + 2. \tag{5}$$

In Eq.3, $R_1$ represents the regret in the "Index Assignment" phase, $R_2$ represents the regret in the "Round Robin" phase, and $R_3$ represents the regret in the "Exploitation" phase. Eq.4 holds based on Lemma 3 and the fact that the "Index Assignment" phase lasts for $N$ time steps. Combining Lemma 4, Lemma 5 and Lemma 8, we conclude that, conditional on $\mathcal{E}$ and $\mathcal{E}^a$, with probability more than $1 - \frac{2}{T}$, player will enter the "Exploitation" phase with optimal stable arm after no more than $r$ rounds in "Round Robin" phase. Thus, Eq.5 holds.

As for arm-pessimal stable regret, we can easily conclude the result according to the fact that: if all players match with their optimal stable arms, then all arms match with their pessimal stable players. $\square$

# C UNKNOWN TIME HORIZON

In this section, we extend the setting where the time horizon $T$ is unknown.

The doubling trick ([Besson and Kaufmann, 2018, Auer et al., 1995]) is a commonly used method to address unknown time horizon $T$ and converses the bound of $O(\log T)$. We adopt the doubling trick both on the total time horizon the exploration.

By using exponential doubling trick, the whole time horizon $T$ is divided into several periods. In every period $r_1$, all players will suppose the time horizon $T_{r_1} = 2^{2^{r_1}}$. When they act more than $T_{r_1}$ time steps in total, they will update their assumption and enter the next period, i.e. suppose $T_{r_1+1} = 2^{2^{r_1+1}}$. The doubling trick will also be used in the exploration. Specifically, the first exploration will last for $2K_2$ time steps, the second exploration lasts for $2 \cdot 2K_2 = 4K_2$ time steps, the third for $2 \cdot 4K_2 = 8K_2$ time steps, and so on.

Moreover, we suppose that arms are also not aware of the time horizon $T$. Thus, they also update their beliefs. Define the event $\mathcal{E}^a(r_1) = \{\forall j \in \mathcal{N}, k \in \mathcal{K}, |\hat{u}_{kj}^a - u_{kj}^a| < 2\sqrt{\frac{2^{r_1}}{N_{kj}^a}}\}$. We say the arms adopt modified $R$ rational method with unknown time horizon, if for every period $r_1$, conditional on $\mathcal{E}^a(r_1)$, after no more than $R\frac{2^{r_1}}{(\Delta^a)^2}$ samples for every player, the arms can estimate their utilities accurately.

---

**Algorithm 9** Round Robin ETC (for a player $j$ with unknown $T$)

---
1: Index $\leftarrow$ *INDEX-ASSIGNMENT$(N, \mathcal{K})$*
2: **for** $r_1 = 1, 2, ...$ **do**
3:     OPT $\leftarrow \emptyset$, $N_2 \leftarrow N, \mathcal{K}_2 \leftarrow \mathcal{K}, r_2 \leftarrow 1$
4:     **while** OPT$= \emptyset$ **do**# when $j$ hasn't found her potential optimal stable arm yet
5:         **for** $t = 1, 2, ..., 2^{r_2}K_2$ **do** # Exploration Sub-Phase
6:             Pull $(\text{Index} + t) \mod K_2 = m$-th arm in $\mathcal{K}_2$, update $\hat{u}_{jk}, N_{jk}, r_2 \leftarrow r_2 + 1$
7:         **end for**
8:         **if** for every $k_1 \neq k_2 \in \mathcal{K}_2$, $\text{UCB}_{jk_1} < \text{LCB}_{jk_2}$ or $\text{LCB}_{jk_1} > \text{UCB}_{jk_2}$ **then**
9:             Success $\leftarrow 1$ # the player achieves a confident estimation
10:         **end if**# Communication Sub-Phase
11:         Success $\leftarrow$ *COMM(Index, Success, , $N_2, K_2, \mathcal{K}_2$)*
    # Update Sub-Phase
12:         OPT $\leftarrow$ *GALE-SHAPLEY*$(\text{Success}, N_2, \mathcal{K}_2, \hat{\boldsymbol{u}}_j, \boldsymbol{N}_j)$
13:         **if** Success$= 1$ **then Break while**
14:         **end if**
15:         **for** $t = 1, ..., N_2K_2$ **do**
16:             **if** $t = (\text{Index} - 1)K_2 + m$ **then**
17:                 Pull arm $k$ that is $m$-th arm in $\mathcal{K}_2$
18:                 **if** $C_j = 1$ **then** $\mathcal{K}_1 \leftarrow \mathcal{K}_1 \setminus \{k\}, N_1 = N_1 - 1$
19:                 **end if**
20:             **end if**
21:         **end for**
22:         $N_2 \leftarrow N_1, \mathcal{K}_2 \leftarrow \mathcal{K}_1$
23:         Index $\leftarrow$ *INDEX-ASSIGNMENT$(N_2, \mathcal{K}_2)$*
24:     **end while**
25:     Pull OPT arm
26: **end for**

---

**Theorem 3.** *If every player runs Algorithm 9, and arms adopt modified strategies that satisfy R-rational condition, then the optimal stable regret of any player $j$ can be upper bounded by:*

$$\overline{R}_j(T) \leq N + \frac{32K(c+2)^2\log T}{\Delta^2} + rN\log(\frac{32K(c+2)^2\log T}{\Delta^2})(KN(N-1) + N + K + 1) + (4KN + 2)r, \quad (6)$$

*where $r = \log\log T + 1$.*

**Theorem 4.** *If every player runs the modified algorithm of Algorithm 1 based on doubling trick on the exploration, and*

*arms adopt R-rational method, then the optimal stable regret of any player $j$ can be upper bounded by* [3]:

$$\overline{R}_j(T) \leq N + \frac{8K(c+2)^2 \log T}{\Delta^2} + N \log(\frac{16K(c+2)^2 \log T}{\Delta^2})(KN(N-1) + N + K + 1) + 4KN + 2. \quad (7)$$

*Proof.* Since doubling trick is only used on the exploration, after $r$ rounds of exploration, every available arm is explored for $2^{r+1} - 2$ time steps. By similar analysis with Lemma 5, we can conclude that conditional on $\mathcal{E}$, after no more than $\frac{8K(c+2)^2 \log T}{\Delta^2}$ times in the exploration, every player will achieve a confident estimation on the available arm set $\mathcal{K}_2$. Then following the proof in Theorem 1, we can simply get this theorem. □

*Proof of Theorem 3.* According to the design of Algorithm 9 and Theorem 4, we can simply get the conclusion by summing regret in each period. □

**Remark 1.** *Similar results for arm regret can be easily obtained due to the fact that: if all players match with their optimal stable arms, then all arms match with their pessimal stable players.*

## D    OMITTED PROOFS IN SECTION 4

In this section, we provide a regret analysis for the collaborative case. Before we prove the main theorem, we provide some lemmas that will be useful.

**Lemma 9.** *If player $j$ obtains successful learning, all participants achieve confident estimations and all players obtain successful learning.*

*Proof.* According to the design of the algorithm, if player $j$ achieves successful learning, she succeeds at the time step corresponding to her index on each arm during communication. It's important to note that arm 1 will choose player $j$ at that time step only if and when all players attain confident estimations, choose arm 1 during the previous check, and arm 1 achieves a confident estimation. Other arms will select player $j$ at the time step corresponding to her index only when they achieve confident estimations. As a result, all participants attain confident estimations. Furthermore, it can be easily concluded that all players achieve successful learning. □

**Lemma 10.** *Conditional on $\mathcal{E}$ and $\mathcal{E}^a$, a participant will achieve a confident estimation after no more than $\lceil \frac{64}{K^2 \Delta_*^2} \rceil$ rounds of exploration, where $\Delta_* = \min\{\Delta, \Delta^a\}$.*

*Proof.* After $\lceil \frac{64}{K^2 \Delta_*^2} \rceil$ rounds of exploration, every arm is matched with each player for at least $\frac{64}{\Delta_*^2} \log T$ time steps, i.e. $N_{jk} \geq \frac{64 \log T}{\Delta_*^2}$ and $N_{kj}^a \geq \frac{64 \log T}{\Delta_*^2}$ hold for every player $j$ and every arm $k$. For $k_1, k_2 \in \mathcal{K}$ that $u_{jk_1} > u_{jk_2}$, conditional on $\mathcal{E}$, we have that:

$$
\begin{aligned}
\text{LCB}_{jk_1} &= \hat{u}_{jk_1} - 2\sqrt{\frac{\log T}{N_{jk_1}}} > u_{jk_1} - 4\sqrt{\frac{\log T}{N_{jk_1}}} \geq \Delta_j + u_{jk_2} - 4\sqrt{\frac{\log T}{N_{jk_1}}} \\
&> \Delta_j + \hat{u}_{jk_2} - 2\sqrt{\frac{\log T}{N_{jk_2}}} - 4\sqrt{\frac{\log T}{N_{jk_1}}} > \Delta_j + \hat{u}_{jk_2} + 2\sqrt{\frac{\log T}{N_{jk_2}}} - 4\sqrt{\frac{\log T}{N_{jk_2}}} - 4\sqrt{\frac{\log T}{N_{jk_1}}} \\
&= \text{UCB}_{jk_2} + \Delta_j - 4\sqrt{\frac{\log T}{N_{jk_2}}} - 4\sqrt{\frac{\log T}{N_{jk_1}}} \\
&\geq \text{UCB}_{jk_2}.
\end{aligned}
$$

Similarly, we can prove that for $j_1, j_2 \in \mathcal{N}$ that $u_{kj_1}^a > u_{kj_2}^a$, conditional on $\mathcal{E}^a$, $\text{LCB}_{kj_1}^a > \text{UCB}_{kj_2}^a$. □

**Lemma 11.** *If a participant achieves a confident estimation, conditional on $\mathcal{E}$ and $\mathcal{E}^a$, the estimation of the participant is correct.*

---

[3]Similar result for arm pessimal stable regret can be simply obtained.

*Proof.* Conditional on $\mathcal{E}$, for any $k_1, k_2 \in \mathcal{K}_2$ that $\text{LCB}_{jk_1} > \text{UCB}_{jk_2}$, have:

$$u_{jk_1} > \text{LCB}_{jk_1} > \text{UCB}_{jk_2} > u_{jk_2}.$$

Thus, the correctness of player $j$'s estimation is proved. The correctness of arms' estimations can be similarly obtained. □

Combining Lemma 9 and Lemma 11 and based on the property of *GALE-SHAPLEY* algorithm, we can conclude that, conditional on $\mathcal{E}$ and $\mathcal{E}^a$, once a player obtains successful learning, she will exploit her optimal stable arm till the end. Together with these lemmas and analysis, we now move to our main theorem.

*Proof.* By decomposing the player optimal stable regret and using the above lemmas, we obtain

$$
\begin{aligned}
\overline{R}_j(T) &= \mathbb{E}[R_1 + R_2 + R_3 | \mathcal{E} \cup \mathcal{E}^a] + T \Pr[\neg \mathcal{E}] + T \Pr[\neg \mathcal{E}^a] & (8) \\
&\leq N + \mathbb{E}[R_2 + R_3 | \mathcal{E} \cup \mathcal{E}^a] + 4KN & (9) \\
&\leq N + K^3 r \lceil \log T \rceil + r(1 + KN) + 4KN. & (10)
\end{aligned}
$$

In Eq.8, $R_1$ represents the regret in the "Index Assignment" procedure, $R_2$ represents the regret caused by the exploration and communication, and $R_3$ represents the regret caused by exploitation. Eq.9 holds based on Lemma 3 and the fact that the "Index Assignment" phase lasts for $N$ time steps. Combining Lemma 9, Lemma 10, and Lemma 11, we conclude that, conditional on $\mathcal{E}$ and $\mathcal{E}^a$, player will exploit optimal stable arm after no more than $r$ rounds of exploration and communication. Thus, Eq.10 holds.

As for arm-pessimal stable regret, we can easily conclude the result according to the fact that: if all players match with their optimal stable arms, then all arms match with their pessimal stable players. □

# E  SIMULATION

In this section, we provide numerical results to show the performance of our algorithm. We estimate the average player-optimal stable regret and standard deviations of regret over 30 independent runs.

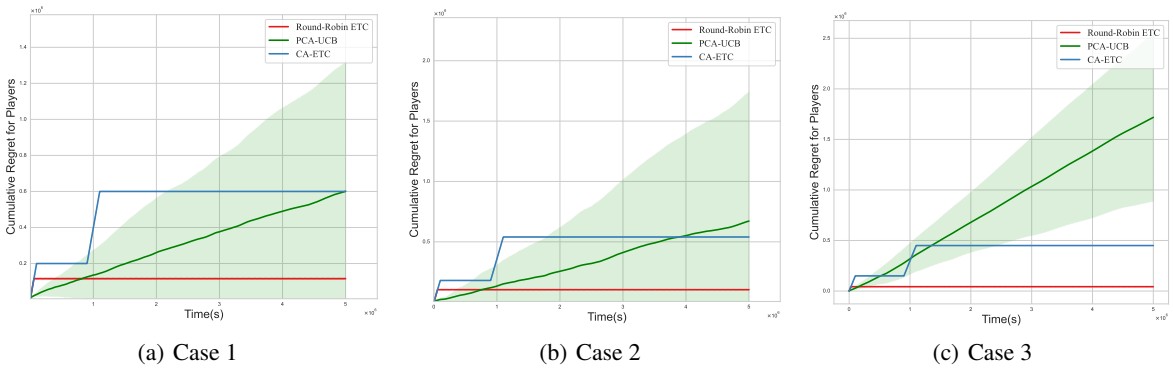

(a) Case 1      (b) Case 2      (c) Case 3

**Baselines.**

- *PCA-UCB* is a conflict-avoiding algorithm with the random delay parameter $\lambda$. This algorithm extends CA-UCB, which only achieves a $O(\log^2 T)$ regret bound compared to player-pessimal regret, even in the one-sided setting. We set $\lambda = 0.9$ based on the simulations in [Pokharel and Das, 2023]. Since Pokharel and Das [2023] do not reveal detailed strategies for the arm side, in our simulations, we assume that the arms choose candidates with the highest UCB.

- *CA-ETC* is a multi-epoch ETC type algorithm that theoretically obtains the regret. Same as the simulation in [Pagare and Ghosh, 2023], we choose $\gamma$, which determines horizon length, to be 0.25. In [Pagare and Ghosh, 2023], the epoch length $T_0$ is chosen based on $\Delta^a$ and $\Delta$. The authors do not disclose the specific details of how $T_0$ is chosen for simulations, but they emphasize that $T_0$ should be optimistically high. Thus, we set $T_0$ to be 50000. Moreover, CA-ETC requires arms to adopt specific symmetric strategies compared to players.

We investigate three scenarios in the context of a multi-armed bandit problem involving five players and five arms in two instances, and four players and four arms in one instance. In the former two cases, the minimum gaps between players and arms are set to 0.2, while in the latter case, the minimum gap is $0.25$. The preferences for these scenarios are described as follows:

(1) Case 1:

$$p_1 : a_4 \succ a_1 \succ a_2 \succ a_3 \succ a_5, \quad a_1 : p_1 \succ^a p_4 \succ^a p_2 \succ^a p_3 \succ^a p_5,$$
$$p_2 : a_5 \succ a_2 \succ a_1 \succ a_3 \succ a_4, \quad a_2 : p_2 \succ^a p_5 \succ^a p_3 \succ^a p_1 \succ^a p_4,$$
$$p_3 : a_3 \succ a_4 \succ a_2 \succ a_5 \succ a_1, \quad a_3 : p_2 \succ^a p_1 \succ^a p_3 \succ^a p_5 \succ^a p_4,$$
$$p_4 : a_2 \succ a_1 \succ a_3 \succ a_5 \succ a_4, \quad a_4 : p_3 \succ^a p_5 \succ^a p_2 \succ^a p_4 \succ^a p_1,$$
$$p_5 : a_1 \succ a_3 \succ a_4 \succ a_2 \succ a_5, \quad a_5 : p_1 \succ^a p_3 \succ^a p_2 \succ^a p_4 \succ^a p_5.$$

(2) Case 2:

$$p_1 : a_4 \succ a_1 \succ a_5 \succ a_2 \succ a_3, \quad a_1 : p_3 \succ^a p_1 \succ^a p_5 \succ^a p_2 \succ^a p_4,$$
$$p_2 : a_5 \succ a_1 \succ a_2 \succ a_4 \succ a_3, \quad a_2 : p_5 \succ^a p_2 \succ^a p_1 \succ^a p_4 \succ^a p_3,$$
$$p_3 : a_2 \succ a_5 \succ a_3 \succ a_1 \succ a_4, \quad a_3 : p_3 \succ^a p_1 \succ^a p_2 \succ^a p_5 \succ^a p_4,$$
$$p_4 : a_5 \succ a_2 \succ a_1 \succ a_3 \succ a_4, \quad a_4 : p_1 \succ^a p_2 \succ^a p_5 \succ^a p_4 \succ^a p_3,$$
$$p_5 : a_3 \succ a_5 \succ a_2 \succ a_4 \succ a_1, \quad a_5 : p_1 \succ^a p_4 \succ^a p_5 \succ^a p_3 \succ^a p_2.$$

(3) Case 3:

$$p_1 : a_2 \succ a_1 \succ a_4 \succ a_3, \quad a_1 : p_2 \succ^a p_1 \succ^a p_4 \succ^a p_3,$$
$$p_2 : a_4 \succ a_1 \succ a_2 \succ a_3, \quad a_2 : p_4 \succ^a p_2 \succ^a p_1 \succ^a p_3$$
$$p_3 : a_3 \succ a_2 \succ a_1 \succ a_4, \quad a_3 : p_1 \succ^a p_3 \succ^a p_4 \succ^a p_2,$$
$$p_4 : a_1 \succ a_2 \succ a_3 \succ a_4, \quad a_4 : p_2 \succ^a p_4 \succ^a p_3 \succ^a p_1.$$

From the figures, we can conclude that round-robin ETC outperforms baselines in all cases. Additionally, the results of round-robin ETC and CA-ETC exhibit greater stability than those of PCA-UCB.

The reason why PCA-UCB performs unstably and fails to obtain sublinear results may be as follows:

Firstly, in different runs of simulations, PCA-UCB may converge to different stable matchings instead of consistently converging to the player-optimal stable matching. This variability in convergence could be a significant challenge, as the player-optimal stable matching is more desirable for players. Furthermore, in [Liu et al., 2021], they illustrate an example where even the centralized UCB cannot achieve sub-linear player-optimal regret.

Secondly, when applying PCA-UCB, players adopt a UCB-type method to choose arms, resulting in insufficient samples for arms to learn their preferences. Consequently, arms may provide inaccurate feedback in the two-sided learning setting, potentially leading to unstable matching or an extended time to convergence.

Regarding CA-ETC, it is important to note that the players persist in exploring arms even after each participant has acquired knowledge of her own preferences. Consequently, regret continues to accumulate over time. The regret curve exhibits a stair-like pattern, reflecting the periodic increments in regret.

Furthermore, analysis of the depicted data indicates a consistent decrease in regret associated with CA-ECT and Round-Robin ETC as both the number of players and arms decreases, and as the minimal gap increases. In contrast, the regret observed for PCA-UCB exhibits an increase. This trend may be attributed to several factors outlined previously: firstly, the tendency to converge towards lower-quality stable matchings as opposed to player-optimal stable matchings; and secondly, the failure to converge and persistently selecting lower-quality arms. Notably, these issues are intricately linked to the preference structure and detailed utilities rather than the scale of the market or the minimal gap.