# OpenReview forum: "Decentralized Two-Sided Bandit Learning in Matching Market"
_auai.org/UAI/2024/Conference — UAI 2024 poster_

### Official Review · Reviewer_f2Fa · 2024-03-03

**Q2-1 Originality-Novelty:** 2
**Q2-2 Correctness-Technical Quality:** 3
**Q2-5 Clarity Of Writing:** 4

**Q10 Ethical Concerns:**

I do not see any potential ethical concerns.

**Q1 Summary And Contributions:**

The paper’s title informatively conveys the topic being studied:
- Matching market: Both sides of the market have preferences over the other side, in contrast to the classical bandit setting where only the players have preferences over arms (assuming a single player is a special case of that)
- Decentralized: Each agent executes the protocol independently and only obtains information through her own matching outcomes.
- Two-sided learning: Both sides must learn their preferences through repeated interactions. On each time step, players propose to arms, and each arm chooses which player to match with.

The primary result is a decentralized protocol with sublinear regret. The specific bound is O(K log T / Delta^2), where K is the number of arms, and Delta is the minimum utility gap any player has between distinct arms. The main assumptions are:
1. Each arm is “rational” in that once it has enough samples from each player, it always matches with the player which provides the highest empirical utility (among the players that proposed to it).
2. The minimum player utility gap (over arms) and the minimum arm utility gap (over players) are within a constant factor.
The authors also state that if the arms are required to adopt particular collaborative policies, Assumption 2 can be removed.

**Q2-3 Extent To Which Claims Are Supported By Evidence:**

3: Good: the main claims are supported by convincing evidence (in the form of adequate experimental evaluation, proofs, (pseudo-)code, references, assumptions).

**Q2-4 Reproducibility:**

3: Good: key resources (e.g. proofs, code, data) are available and key details (e.g. proofs, experimental setup) are sufficiently well-described for competent researchers to confidently reproduce the main results.

**Q3 Main Strengths:**

The problem is natural and well-motivated, and the result is strong. The writing and organization are also quite good. For example, Table 1 is very useful in understanding the comparisons with prior work. I also appreciated that the authors discussed the significance of the assumptions that they do and don’t make (e.g., the paragraphs “No Central Communicator.” and “No Observation of Winning Players” in Section 2.1). The authors also provide an overview of the challenges they faced and the solutions they found (Section 3.1). I would love for more papers to include these sorts of discussions.

**Q4 Main Weakness:**

The primary weakness in my opinion is limited novelty when compared to Pagare and Ghosh (2023). PG23 study essentially the same problem; indeed, their title is almost the same (“Two-Sided Bandit Learning in Fully-Decentralized Matching Markets”). However, the authors of this submission dedicate only two sentences to comparing with PG23. Their main statement is “However, [PG23’s] algorithm requires a hyper-parameter T_0, which is subject to certain constraints related to some knowledge of the minimal gap [Delta], and requires arms to adopt a symmetry algorithm akin to players”.

After reading PG23 myself, this is my understanding of the differences:
1. PG23 assume that the arms adopt a specific policy, while this submission allows arms to follow any policy which is “rational” in the sense defined above.
2. PG23 do not need the utility gap assumption
3. It is true that PG23’s algorithm has a parameter T_0 which must satisfy an inequality depending on Delta, so in that sense their algorithm needs some knowledge of Delta. However, the inequality is quite mild: in particular, for fixed Delta, the inequality is trivially satisfied for large T.
4. This submission has a better regret bound when Delta is constant (logarithmic in T, while PG23’s is polynomial in T).

Note that #4 was not mentioned explicitly by the authors; I determined this from Table 1 (which, again, is super useful.)

My evaluation scores are based on the assumption that my summary above is correct. Given that, I think that this submission retains some novelty, but the novelty is is somewhat limited when compared to PG23. If my summary is not correct, I would appreciate if the authors could correct me and clarify the precise differences.

I would recommend that in future drafts, the authors provide a much more thorough comparison with PG23 so that reviewers are in a better position to judge novelty. Even if my summary is correct, I think this work still has enough novelty to be publishable if a more thorough comparison with PG23 is provided. I also wonder if #4 should be emphasized in the text

**Q5 Detailed Comments To The Authors:**

(Some of these comments are phrased as questions, but I don’t expect responses. Feel free to use or discard this feedback as you wish.)

- Pg 2: The phrasing “assumption of sex-wide homogeneity” implies a specific motivating example (heterosexual dating). Consider a more general phrasing like “assumption that each side of the market is homogeneous”.
- Table 1 is super helpful. It might be worth clarifying that “one-sided” vs “two-sided” is about whether the *learning* is two-sided, not the market (I believe all the markets are two-sided)
- Do you really need N <= K? Is this so that you can do round robin with no conflicts between players? Would it not be sufficient to have a rotating set of N - K players not participate each round in the exploration phase? It might be worth mentioning this in the paper, even in a footnote
- Pg 3, left column: should the assumption be “D Delta^a >= Delta” instead of “D Delta^a >= Delta_j”?
- Bottom of pg 8, left column: the end-of-proof box overlaps with the footnote
- In Section 4, consider referencing which specific section of the appendix contains the details?

**Q9 Complying With Reviewing Instructions:**

Yes

---

> ### Author Rebuttal · Authors · 2024-04-05
>
> Thanks for your valuable comments. We will incorporate your suggestions to further polish our paper in our revision.
>
> Weakness. We respectfully disagree with your assertion regarding limited novelty. Let $\Delta^*$ denote the minimal gap for players and arms, defined as $\Delta^*=\min\set{\Delta,\Delta^a}$. Firstly, we apologize for the notation typo in our main text; PG23's algorithm relies on $\Delta^*$ rather than $\Delta$. Secondly, regarding the differences, not all are well justified, particularly with regards to points #2 and #3. We will address your understanding of these differences below:
>
> \#1. YES. Assuming a specific arm policy is indeed a strong assumption, as it allows arms to strategically aid players without optimizing their rewards. Furthermore, our paper proposes an additional algorithm (in Appendix D) that does not require the arm gap assumption, but with arms adopting specific policies as an extension. This algorithm achieves significantly smaller regret compared to PG23's.
>
> \#2 and \#3. NO. PG23's algorithm includes a parameter that must satisfy an inequality dependent on $\Delta^*$. This means that either (1) know a lower bound $d$ of $\Delta^*$, i.e., $d\leq \Delta^*$, or (2) their algorithm can only solve instances where $\Delta^*$ is not very small. Case (1) constitutes a very strong assumption in matching bandits, as with this assumption, even a simple ETC algorithm can achieve sub-linear or even logarithmic regret. Case (2) imposes far stricter assumptions on utility gap than ours. In our case, we only require a ratio in learning difficulty between arm and player sides, rather than restrictions on the difficulty of learning itself.  However, in PG23, they require the learning difficulty cannot be very high. Consequently, our algorithm can effectively address their instances, whereas theirs is unable to resolve ours. For example, their algorithm may only be able to solve the case where $\Delta^*\ge\frac{1}{100}$. In contrast, our algorithm can solve the case where $\Delta\ge\frac{1}{100}\Delta^a$ , including cases where $\Delta^*\ge\frac{1}{100}$ (note that $\Delta\le 1$) and $\Delta^*=\frac{1}{10000}=\Delta=\Delta^a$.
>
>
> \#4. YES. In matching bandits, we always consider instance-dependent regret. Our regret bound significantly outperforms theirs, from $O(T^\gamma)$ to $(\log T)$. Additionally, our regret bound matches the state-of-the-art result in the simpler one-sided learning case, with no restrictions on observation or special preference structure. Moreover, our regret bound is also tight in terms of $T$ and $\Delta$.
>
> Thirdly, the technical ideas of our algorithm and PG23's are significantly different. For instance, in PG23, PG23's main approach involves multi-epoch ETC, leading to over exploration and high regret. In contrast, our method employs techniques like deliberate conflicts to end the exploration as soon as possible.
>
> Lastly, we summarize all the differences below. We will also add more discussion about the differences between our work and PG23's in our revision.
>
> | Comparisons    |PG23's |Round-Robin ETC|Our Extension|
> |----- |:--------:|:-----------:|:----------:|
> |Arm Policy  |specific policy  | any rational policies | specific policy|
> |Gap Assumption| not small gap or a known lower bound of minimal gap  (the strongest assumption)| comparable ratio between two sides  (no restrictive assumption)| NO |
> |Regret Bound| $O(T^\gamma)$($\gamma \in (0,1)$)|$O(\log T)$|$O(\log T)$|
> |Technical Idea| multi-epoch ETC, over exploration | deliberate conflicts, end the exploration as soon as possible| deliberate conflicts to communicate on both sides, end the exploration as soon as possible|
> |Main Contribution| sub-linear regret in two-sided learning setting|1. significant improved regret bound (logarithmic regret) 2. boarder range of arm policies and less restrictive assumption on gap| 1.  significant improved regret bound (logarithmic regret) 2. no assumption on gap|
>
> Q3. In the case where $N>K$, there will be unmatched players in stable matching. Previous works typically assume $N \leq K$ to ensure that every player can match with one arm.  We and almost all previous work do not allow players to choose no arm. However, by simply modifying the exploration phase, our algorithm can still be effective in the case where $N>K$, with regret no more than  $O(N\log T/\Delta^2)$. Thank you for bringing this to our attention; we will include this in our revision.
>
> Minor Questions.
> Thanks for pointing these out, we will address them in the revision.

---

### Official Review · Reviewer_VSeq · 2024-03-19

**Q2-1 Originality-Novelty:** 3
**Q2-2 Correctness-Technical Quality:** 2
**Q2-5 Clarity Of Writing:** 3

**Q10 Ethical Concerns:**

No. It will be interesting to see if certain players will benefit more (lower regret or lower computational complexity) than others using this algorithm though.

**Q1 Summary And Contributions:**

This paper studies the two-sided bandit learning in a matching market in a fully decentralized setting, which does not require full observation and special preference structure assumptions. The authors proposed an algorithm Round-Robin ETC and theoretically showed that it achieves a "Rational Condition" on the arms' side (passive) and O(log T) stable regret on the players' side (proactive). This regret bound, according to the authors, achieves state-of-the-art performance on the players' side in this field.

**Q2-3 Extent To Which Claims Are Supported By Evidence:**

2: Fair: the main claims are somewhat supported by evidence (but the experimental evaluation may be weak, or does not match entirely with the claims, important baselines may be missing, proofs contain important ideas but lack rigor, algorithmic details are only discussed superficially, references are imprecise, assumptions are not sufficiently motivated or explicated, etc.).

**Q2-4 Reproducibility:**

3: Good: key resources (e.g. proofs, code, data) are available and key details (e.g. proofs, experimental setup) are sufficiently well-described for competent researchers to confidently reproduce the main results.

**Q3 Main Strengths:**

1. The overall flow of the paper is easy to follow and the main results and discussions are presented in a clear way
2. The comparison in Table 1 is informative

**Q4 Main Weakness:**

1. The intuition behind why the Round-Robin ETC can outperform previous works is not obviously stated
2. The existence of a stable matching is not discussed in the main article, which is a necessity for the regret calculations
3. The incentives for players to participate in the Round-Robin ETC and follow the procedures are not sufficiently discussed, e.g., what can be the outside options, and why is it rational for players to collaboratively follow the steps in the algorithms?
4. The numerical results are not included in the main article
5. The computational complexity of this algorithm compared with previous works is not discussed, and in situations where the number of pulling is limited, can this algorithm still outperform the baselines?

**Q5 Detailed Comments To The Authors:**

Please address the weaknesses listed above. Additionally, I have the following questions/points:

1. When the assumptions in previous works, e.g., full observation, and specific preference structures are added to this work, how much improvement does each of them bring? If no improvements, why?
2. Why are the numerical results not in the main article but in the appendix?
3. Why is the arm-side performance missing in Table 1?

Minor points:
1. Page 4, right column, line 2, Gale and Shapley repeated twice
2. Please consider adding a notation table in the appendix

**Q9 Complying With Reviewing Instructions:**

Yes

---

> ### Author Rebuttal · Authors · 2024-04-05
>
> Thank you for your valuable comments. We will incorporate your suggestions to further refine our paper in the upcoming revision.
>
> W1. In our paper, we outline the challenges encountered in the two-sided learning scenario and illustrate how our techniques overcome them. For instance, we employ deliberate conflicts to facilitate player cooperation in the absence of observation and a central communicator. This is a key factor contributing to the success of our algorithm. Compared to previous work, such as PG23’s algorithm, which adopts a multi-epoch approach leading to over-exploration, our algorithm incorporates communication and confidence bounds to help players enter the exploitation phase as soon as possible.
>
> W2. The existence of stable matching is guaranteed by the GS algorithm (Gale and Shapley, 1962). We will include this clarification in our revision.
>
> W3. The outside option for players is to deviate from our algorithm. Previous studies (e.g., Boursier and Perchet, 2020) typically employ $\epsilon$-NE to demonstrate algorithm robustness, where $\epsilon$ scales with $O(\log T)$ or $o(T)$. We argue that players lack a clear incentive to deviate from our algorithm, as it ensures matching with their optimal stable arms with low regret. Deviating from our algorithm would not significantly benefit them, considering they have already achieved $O(\log T)$ regret. Additionally, the rationale for communication has been discussed in our paper.
>
> W4 & Q2. Since our focus is primarily on theoretical results, experiments are provided in the Appendix due to space constraints.
>
> W5. In the study of matching bandits, to the best of our knowledge, no prior works have compared or calculated the computational complexity of their algorithms. When the number of pulls is limited, players may be unable to learn their own preferences, resulting in scenarios where no algorithm can achieve sub-linear regret.
>
> Q1. Regarding our regret bound, it is tight in terms of $\Delta$ and $T$, and from our perspective, cannot be improved order-wise. However, when full observation and specific preference structures are allowed, overheads can be reduced. This is because more effective communication becomes feasible under such conditions. Nonetheless, the exploration component, which pertains to learning one's own preferences, remains a significant contributor to regret and cannot be improved.
>
> Q3. Firstly, addressing the issue of arm-side regret, previous works lack theoretical guarantees in this regard, primarily focusing on player-side regret. Secondly, as stated, in our algorithm, players transition to the exploitation phase, pulling their optimal stable arms. Consequently, arms will match with their stable pairs during the exploitation phase, resulting in arm-side regret of $O(\log T)$.
>
> Minor Points:
> Thanks for highlighting these. We will address them accordingly in our revision.

---

### Official Review · Reviewer_CcJF · 2024-03-22

**Q2-1 Originality-Novelty:** 3
**Q2-2 Correctness-Technical Quality:** 3
**Q2-5 Clarity Of Writing:** 3

**Q10 Ethical Concerns:**

No ethical concerns.

**Q1 Summary And Contributions:**

The paper studies decentralized two-sided matching market with unknow preference and only bandit feedbacks. The paper proposed a ROUND-ROBIN ETC learning algorithm and proved that the algorithm can achieve a $O(K log T/ \Delta^2)$ regret.

**Q2-3 Extent To Which Claims Are Supported By Evidence:**

4: Excellent: all claims are supported by very convincing evidence (in the form of comprehensive experimental evaluation, rigorous mathematical proofs, detailed (pseudo-)code, precise references, well-motivated and realistic assumptions) and the authors deliver what they promise.

**Q2-4 Reproducibility:**

4: Excellent: key resources (e.g. proofs, code, data) are available and key details (e.g. proof sketches, experimental setup) are comprehensively described for competent researchers to confidently and easily reproduce the main results.

**Q3 Main Strengths:**

The decentralized two-sided matching market with unknow preference is a very interesting problem and compared to the literature, the authors make less assumptions and proved that the proposed algorithm has logarithmic regret bound that has the same order
as the state-of-the-art guarantee in the simpler one-sided learning setting.

**Q4 Main Weakness:**

My major concern is the algorithm part. The communication part of the algorithm is not completely clear to me although I understand the intuition. Even Example 1 does not mention the details of the communication phase of the proposed algorithm. I would suggest the authors use some figures or include more details in the example.

**Q5 Detailed Comments To The Authors:**

Other comments:

1. What is the arm-side regret using the proposed algorithm?

2. The authors mentioned that "This regret bound is tight in terms of T and $\Delta$" but I cannot find any proof in the paper. Is there any lower bound in the literature with the same order?

**Q9 Complying With Reviewing Instructions:**

Yes

---

> ### Author Rebuttal · Authors · 2024-04-05
>
> Thanks for your valuable comments. We will incorporate your suggestions to further polish our paper in our revision.
>
> Weakness. We will incorporate figures and more details in the example and polish our algorithm part.  In Example 1, the round-robin phase comprises three rounds, resulting in two rounds of communication.
>
> In the first round, only player $p_3$ attains confident estimations. Consequently, during communication, player $p_1$ will initiate by pulling the three arms in order twice, acting as the transmitter. Subsequently, player $p_2$ will sequentially pull the three arms to receive information during the first three pulls, serving as the receiver. In the subsequent three pulls, player $p_3$ will serve as the receiver, pulling the three arms to obtain information. However, when player $p_3$ pulls arms $1$ and $2$, she will get rejected, as these arms prefer player $p_1$. Consequently, no player achieves successful learning in this round.
>
> In the second round, players $p_1$ and $p_3$ achieve confident estimations. During this round of communication, since players $p_1$ and $p_3$ have achieved confident estimations, they will not pull the same arm as the receiver when transmitting information. Consequently, receivers will not face rejection when receiving their information. Player $p_2$ will pull the same arm as the receiver. However, since players $p_1$ and $p_3$ are preferred over player $p_2$ on all arms, they will also avoid rejection when receiving information from $p_2$. This implies that players $p_1$ and $p_3$ obtain successful learning and will proceed to the exploitation phase.
>
> In the final round, with only one player remaining, there is no communication.
>
> Q1: In our algorithm, all players transition to the exploitation phase, selecting their optimal stable arms. Consequently, arms are matched with their stable pairs during exploitation, resulting in arm-side regret of $O(\log T)$. We will include this clarification in our revision.
>
> Q2: In the simpler one-sided setting, there exists a lower bound of $O(\log T/\Delta^2)$ (Sankararaman et al., 2021), even under the special case where arms exhibit identical and known preferences (globally rank). If arms consistently select the optimal candidate (a rational strategy), our setting reduces to the one-sided learning scenario. Therefore, this lower bound also applies to the two-sided learning setting, and our regret bound aligns with this lower bound.

---

### Official Review · Reviewer_3iDn · 2024-03-28

**Q2-1 Originality-Novelty:** 3
**Q2-2 Correctness-Technical Quality:** 3
**Q2-5 Clarity Of Writing:** 3

**Q1 Summary And Contributions:**

The authors consider the two sided matching market problem in the case of repeated rounds and decentralized communication settings. Regret bounds are provided for the player side and obtain and are tight. Experiments are provided in the supplementary section.

Overall: The paper is easy to read and the main algorithms and results are provided. Experiments are scanty and some questions remain.

**Q2-3 Extent To Which Claims Are Supported By Evidence:**

3: Good: the main claims are supported by convincing evidence (in the form of adequate experimental evaluation, proofs, (pseudo-)code, references, assumptions).

**Q2-4 Reproducibility:**

3: Good: key resources (e.g. proofs, code, data) are available and key details (e.g. proofs, experimental setup) are sufficiently well-described for competent researchers to confidently reproduce the main results.

**Q3 Main Strengths:**

(i) The assumptions are much weaker than prior work such as less information is provided to the players as to who wins each round but are also a better  model of real-world scenarios.
(ii) A working example showcases the various stages of the proposed algorithm as the players enter round-robin, exploration, and finally exploitation rounds.

**Q4 Main Weakness:**

(a) Communication constraints seem to be an important factor in the setup. No bounds are provided on the number of communication bits in the decentralized settings over the prior art.
(b) Experiments were not conducted on a variety of players, arm sizes and/or gaps delta to validate the regret scaling in the bounds.

**Q5 Detailed Comments To The Authors:**

(1) N does not appear in the regret terms. Is that due to being absorbed (N<K)? What happens to the algorithm when N > K empirically?
(2) Also, what happens if the arms have a capacity of more than one and can match with p players at each round?
(3) Is there a contextual version of this matching? It seems like the arms and players should have feature vectors which help define their reward function?
(4) Can we quantify the amount of communcation between the players over T rounds?

**Q9 Complying With Reviewing Instructions:**

Yes

---

> ### Author Rebuttal · Authors · 2024-04-05
>
> Thanks for your valuable comments. We will incorporate your suggestions to further polish our paper in our revision.
>
> W1: Previous works on two-sided learning do not incorporate communication; instead, (Pokharel and Das, 2023) rely on observation to gather more information. In contrast, our approach involves players using deliberate conflicts to communicate. The design of our algorithm ensures that the length of communication is small. (Please also refer to our response to Q4)
>
> W2: Similar to previous studies (e.g., Pagare and Ghosh, 2023), we consider a scenario with $5$ players and $5$ arms. We will include additional simulations in our revision.
>
> Q1: The reason why $N$ does not appear in the regret terms may be due to its absorption ($N<K$). In cases where the number of players $N$ exceeds the number of arms $K$, there will be unmatched players in stable matching. Previous works typically assume $N \leq K$ to ensure that every player can match with one arm. However, when considering the case $N > K$, a similar algorithm can still be effective, with regret no more than $O(N\log T/\Delta^2)$.
>
> Q2: Thank you for pointing out this intriguing avenue for future research. Previous studies referenced in our paper focus on scenarios where arms can only choose one player. If arms have multiple capacities, the strategies of arms need to be reconsidered, and the entire setting may differ significantly. However, we believe that key concepts from our work, such as round-robin exploration and communication strategies, could be adapted for such scenarios.
>
> Q3: While we appreciate the suggestion, our current focus is on addressing the challenges within the existing framework of matching bandits without considering contextual structure. Incorporating contextual features into the matching process would indeed be an interesting avenue for future research. However, it would significantly alter the problem setup and introduce additional complexities. There are existing works that consider contextual linear bandits in matching markets. However, most of them study matching markets from different aspects, e.g., assuming a central platform that can observe the utility of every participant to determine the matching.
>
> Q4: The total amount of communication in our algorithm is small. Roughly estimating, there is an upper bound of $O(N^2/\Delta^2)$ time steps of communication. However, this is a conservative upper bound, as there may be cases where players quickly learn their preferences, requiring minimal communication.

---

### Meta-Review · Area_Chair_ccSC · 2024-04-15

Four reviews have been obtained. One recommends acceptance, and one recommends weak acceptance. Also, one reviewer is not very confident and recommends borderline acceptance. The most detailed review is by Reviewer f2Fa, who recommends borderline rejection and is quite confident. After reading the discussions and looking at the paper on my own, I agree that the paper has limited novelty over the work PG23. In other words, the structure of the proposed algorithms is more or less the same, except that the authors of this paper introduced two sub-oracles in their algorithm to improve the regret bound under some weaker assumptions. The main contribution of this paper is to improve the regret bound from $O(T^{\gamma})$ to $O(\log T)$. In summary, the paper's novelty is limited. However, the analysis is more detailed at the cost of a more complex decentralized algorithm with more communication to improve the regret bound.